# Reactivation of associative structure specific outcome responses during prospective evaluation in reward-based choices

Maya Zhe Wang[1] & Benjamin Y. Hayden[1]

Before making a reward-based choice, we must evaluate each option. Some theories propose that prospective evaluation involves a reactivation of the neural response to the outcome. Others propose that it calls upon a response pattern that is specific to each underlying associative structure. We hypothesize that these views are reconcilable: during prospective evaluation, offers reactivate neural responses to outcomes that are unique to each associative structure; when the outcome occurs, this pattern is activated, simultaneously, with a general response to the reward. We recorded single-units from macaque orbitofrontal cortex (Area 13) in a riskless choice task with interleaved described and experienced offer trials. Here we report that neural activations to offers and their outcomes overlap, as do neural activations to the outcomes on the two trial types. Neural activations to experienced and described offers are unrelated even though they predict the same outcomes. Our reactivation theory parsimoniously explains these results.

[1] Department of Brain and Cognitive Sciences and Center for Visual Science, University of Rochester, Rochester, New York 14627, USA. Correspondence and requests for materials should be addressed to M.Z.W. (email: mwang@mail.bcs.rochester.edu).

Reward-based choices pervade our lives and range from whether to get a cup of tea instead of coffee to whether to become an organ donor. To choose effectively, we must evaluate the potential consequences of our choices in light of the presented options[1,2]. Sometimes these prospective evaluations are based on descriptions, such as when choosing a cupcake based on the menu at a newly opened bakery. Other times, these prospective evaluations are based directly on experience, such as when deciding to have a second cupcake based on how the first one tasted. In both cases, choosing requires generating a prediction about the value of each option, which in turn requires us to mentally link these external options with representations of their outcomes.

Building the mental link between options and outcomes relies on successful encoding of associative structures. That is, it requires us to represent the simple stimulus-outcome/action-outcome associations and/or the more complex associative event sequences that comprise a world model[3]. A good deal of work indicates that the orbitofrontal cortex (OFC) is a key site for representation of associative structures[2,4,5]. Indeed, a recent integrative theory of OFC function suggests that its central role is to instantiate a cognitive map of task space, meaning that it represents the associative structures that are relevant to solving the current task[6,7]. This idea is supported by recent results from lesion studies[3,8,9]. However, the dearth of physiological evidence supporting these ideas limits our understanding of how the encoding of associative structures in OFC contributes to economic choices.

Here we consider two broad possibilities. One possibility would be that reward-predicting offers activate OFC neurons in the same way that the reward itself does. The brain would thus presumably be directly simulating the experience of receiving the reward by replaying the neural response pattern associated with its receipt. In this case, neural response to rewards and to any cues that predict the same reward would be identical. Another possibility would be for the brain to have a distinct pattern of neural response for each unique underlying associative structure. In this case, neural responses to two associative event sequences predicting the same reward would not necessarily overlap.

There is good neural evidence in support of both possibilities. During prospective evaluation, hemodynamic responses in OFC show reactivation of outcome related multi-voxel patterns during the presentation of reward predictive cues[10–12]. OFC neurons also show similar responses to different cues predicting subjectively equally preferred outcomes[13]. Furthermore, OFC shows reactivation of the same set of neurons encoding the outcome when the corresponding offer occurs[14,15]. Other evidence suggests that OFC recruits responses that are unique to each offer-outcome associative event sequence when offers are presented. In one task, each unique associative event sequence (a visual stimulus, an action, and an outcome cue) led to a high or low reward state. After seeing the visual stimulus, participants freely chose and performed one of two actions to complete the sequence that led to the desired reward. The reward states predicted by each sequence were decodable during stimulus presentation and action execution in human central OFC, suggesting that the reward information was represented based on the unique underlying associative structure[16]. Farovik and colleagues[17] demonstrated that OFC ensembles in rats adopted uncorrelated coding schemes when different object-context pairs led to the same reward. Likewise, Tsujimoto and colleagues showed that distinct subsets of macaque OFC neurons encoded the water reward of equal size when it was presented via two routes, as an instruction for choice strategy (stay/switch) versus as a feedback for correct execution of a choice strategy (presumably reflecting distinct associative structures)[18].

Although the two sets of studies may seem to be contradictory, we believe that they can be reconciled. Specifically, we hypothesize that OFC encodes the associative structure specific, and simultaneously, a generic reward signal. During prospective evaluation, only the associative structure specific neural response is present; during retrospective evaluation (that is, immediately after the reward), the associative structure specific neural response is co-activated along with the general reward representation. On the basis of this hypothesis, we predict that when offers are made, neural responses to outcome receipt will be partially reactivated due to the overlapped representation of associative structures. However, offers presented with distinct associative event sequences (here, described and experienced offers) will elicit non-overlapping neural responses, even though they predict the same rewards. Finally, when the choice is made and reward is given, the associative structure specific and the reward general responses will be activated simultaneously. Therefore, we predict that responses to the two outcomes will show partial overlap (Fig. 1c). Here we record single-unit activities from macaque OFC (Area 13) in a riskless reward-based choice task. We report that neural activations to offers and their outcomes overlap, as do neural activations to the outcomes on the two trial types. Neural activations to experienced and described offers are unrelated even though they predict the same outcomes. These results indicate that OFC (Area 13) recruits associative structure specific neural activations to outcomes during prospective evaluation.

## Results

**Behaviour**. On each trial of the choice task, subjects (two male Macaca mulatta) chose between two riskless options, offer 1 and 2, presented on the left and the right side of the screen (Fig. 1a). First, offer 1 cue was presented as a rectangle. On described trials, offer 1 size was revealed by paring the offer 1 cue with one of five coloured rectangles that each was stably associated with a specific reward size. On experienced trials, offer 1 size was revealed by directly paring the offer 1 cue with a water aliquot of one of the same five reward sizes. On both trials types, the size of offer 2 was indicated by one of three other photographic images, each associated with a specific reward size. Trial types, offer positions, and offer sizes were all randomized independently for each trial.

Subjects understood the task well. They chose the option with greater or equal water amount 85.02% of the time (subject H: 88.37%; subject B: 82.45%). This performance was significantly higher than chance level (that is, 56.67%—see Methods; $\chi^2 = 6,166.80$; $P < 0.001$; $n = 31,699$; effect size = 4.34; chi-square test; see Methods). Subjects chose the larger option more often in experienced trials (88.29%) than in described trials (81.72%; $\chi^2 = 268.1$; $P < 0.001$; $n_{experienced} = 15,914$; $n_{described} = 15,785$; effect size = 1.69; chi-square test; Fig. 2). Subjects chose offer 1 more often than expected by optimal strategy: they chose it 44.31% of the time (even though its value was matched to or better than offer 2 only 40% of the time; $\chi^2 = 120.77$; $P < 0.001$; $n = 31,699$; effect size = 1.19; chi-square test). This preference for offer 1 was observed in both described and experienced trials, but was slightly stronger for described than for experienced offers (Fig. 2).

**Neural encoding of offer 1 and outcome amount**. We collected data from 125 neurons in Area 13 of OFC ($n = 65$ in subject H and $n = 60$ in subject B; Fig. 1b, Supplementary Fig. 1, and Methods). Responses of two illustrative neurons are shown in Fig. 3a,b (also see Supplementary Fig. 2a,b). The firing rate of cell #69 during the offer epoch was higher in response to smaller offers than to larger ones in the described trial (B = − 0.003; $P = 0.006$; $n = 231$; $R^2 = 0.03$; linear regression, see Methods).

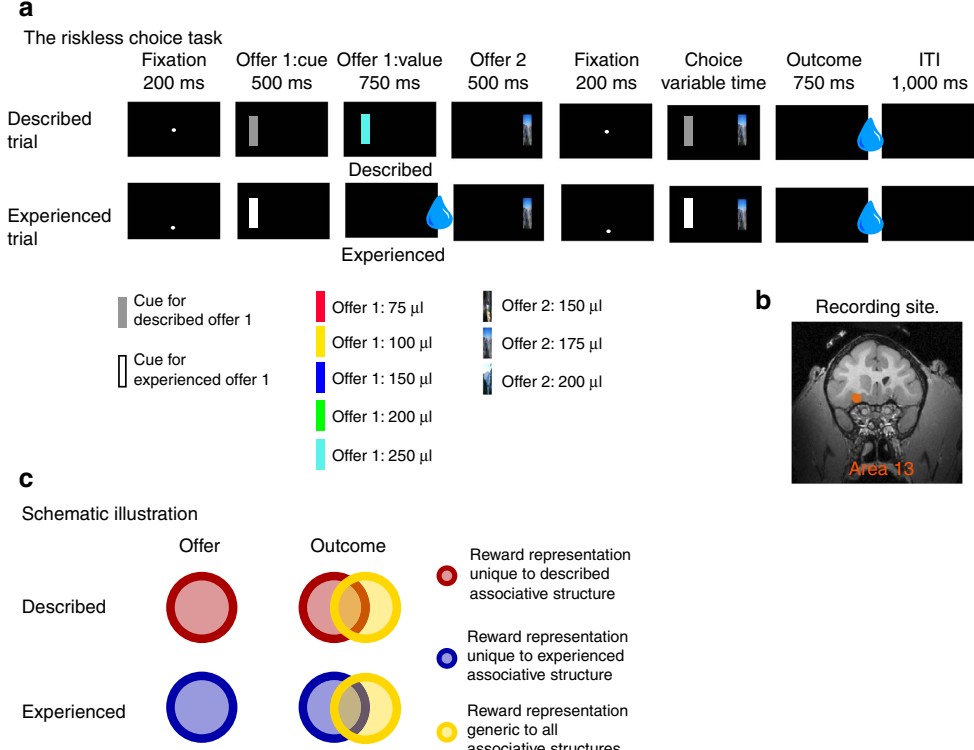

**Figure 1 | Summary of methods and hypothesis.** (**a**) The riskless choice task. Following fixation, the offer 1 cue indicated a described or experienced offer 1. The size of the offer was indicated by coloured rectangle (described trials) or a water aliquot (experienced trials). Presentation of offer 2 followed, and, after a fixation, subject chose by shifting gaze to one of the offer positions. (**b**) Recording site (subject H shown). Area 13 of OFC is highlighted in orange. (**c**) Schematic illustration of our hypothesis. Outcome responses specific to different associative structures were reactivated during prospective evaluation but general reward response insensitive to preceding associative structures was only present during reward delivery after the choice.

During the same epoch, the firing rate of cell #123 was higher in response to larger offers than to smaller ones in experienced trial ($B = 0.004$; $P = 0.003$; $n = 212$; $R^2 = 0.04$; linear regression).

During the offer 1 epoch, the size of the described offer affected firing rate in 12% of neurons ($n = 15/125$; linear regression; Fig. 3c). This proportion is greater than what would be expected by chance ($P = 0.002$; $n = 125$; effect size = 2.4; binomial test). Among these neurons, 53.3% ($n = 8/15$) encoded described offer with positive sign (this proportion is not biased; $\chi^2 < 0.0001$; $P = 0.5$; $n = 15$; effect size = 1.31; chi-square test). The size of the experienced offer affected firing rate in the offer 1 epoch in 16.8% of neurons ($n = 21/125$; Fig. 3c). This proportion is greater than what would be expected by chance ($P < 0.001$; $n = 125$; effect size = 3.36; binomial test). Among experienced offer size-sensitive neurons, 66.7% ($n = 14/21$) encoded experienced offer with positive sign (this proportion is positively biased; $\chi^2 = 3.43$; $P = 0.032$; $n = 21$; effect size = 4.00; chi-square test).

The size of the outcome affected firing rate during the outcome epoch in 9.6% of neurons ($n = 12/125$; linear regression; see Methods) in described trials. This proportion is greater than chance ($P = 0.036$; $n = 125$; effect size = 1.92; binomial test). Among these neurons, 75.00% ($n = 9/12$) encoded outcomes with negative sign (this proportion is negatively biased; $\chi^2 = 4.17$; $P = 0.021$; $n = 12$; effect size = 9.00; chi-square test). The size of the outcome affected firing rate during the outcome epoch in 12.8% of neurons ($n = 16/125$) in experienced trials. This proportion is greater than chance ($P < 0.001$; $n = 125$; effect size = 2.56; binomial test). Among these neurons, 62.50% ($n = 10/16$) encoded outcomes with negative sign (this proportion is not biased; $\chi^2 = 1.13$; $P = 0.144$; $n = 16$; effect size = 2.78; chi-square test).

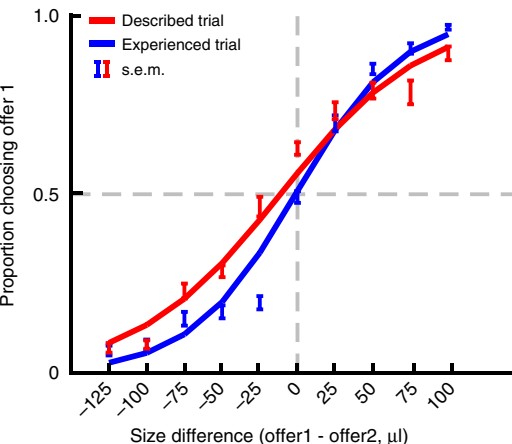

**Figure 2 | Behavior.** Probability of choosing offer 1 as a function of value difference between offer 1 and 2. Preference curves were roughly sigmoidal. Subjects generally chose the higher value offer. Performance was slightly but significantly more optimal (that is, reward-maximizing) for experienced than for described offers.

We saw no evidence that offer 1 encoding was stronger in experienced than described trials (though we might have expected such a pattern due to the higher reward expectations on experienced trials). First, the effect size, as measured by squared coefficients of a linear regression on normalized firing rates against offer 1 size, was not statistically different between described and experienced trials ($t = -0.162$; $P = 0.87$; $n = 125$; effect size = $-0.02$; t-test). Second, the proportions of neurons

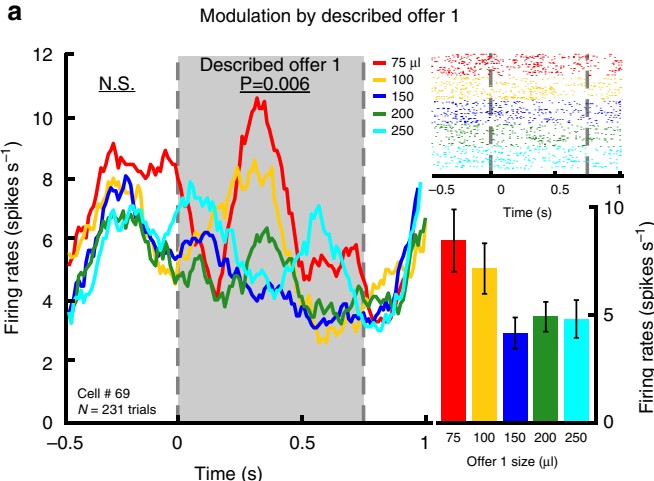

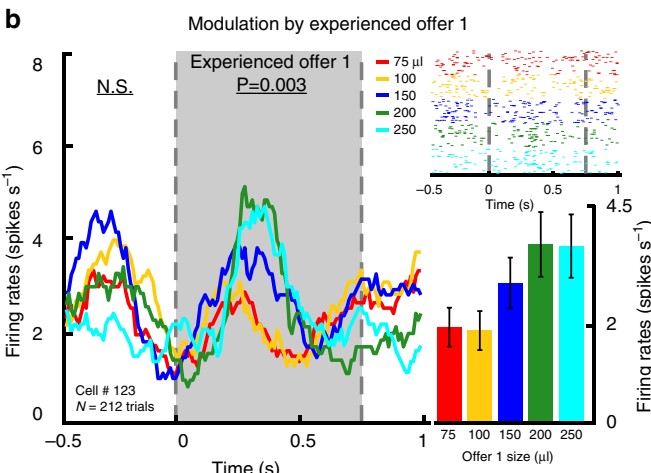

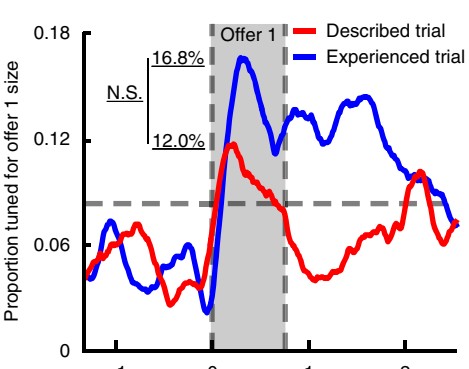

**Figure 3 | Neural encoding of offer.** (**a**) Left: Peristimulus time histogram (PSTH) for one example neuron that showed reduced firing rates in response to larger described offer. Top right: raster plot of the same example neuron; trials are sorted by offer 1 size. Bottom right: bar graph of the same example neuron showing averaged firing rates for each offer 1 size. (**b**) Another example neuron that showed enhanced firing rates in response to larger experienced offer. (**c**) Proportion of neurons in our data set that were selective for the size of described (red) and experienced (blue) offers; sliding-window regression analysis. Chance level is based on binomial test and is indicated by grey dashed line.

tuned for described and experienced offer 1 s were not significantly different ($\chi^2 = 0.81$; $P = 0.36$; $n = 125$; effect size $= 0.68$; chi-square test).

Similarly, we observed no difference in neural responses to experienced offers and outcomes on experienced trials. First, the effect sizes of the offer 1 and outcome responses in experienced trials were not statistically different ($t = 0.98$; $P = 0.33$; $n = 125$; effect size $= 0.09$; $t$-test). Second, the proportion of neurons tuned for offer 1 and outcome in experienced trials were not significantly different ($\chi^2 = 0.51$; $P = 0.48$; $n = 125$; effect size $= 1.38$; chi-square test). Third, the effect sizes of the outcome responses in described and experienced trials were not statistically different across trial types ($t = -0.66$; $P = 0.51$; $n = 125$; effect size $= -0.08$; $t$-test). Thus we observed no neural evidence of diminished marginal utility[19] of outcome (that is, difference in neural response to offer 1 and its corresponding outcome) in experienced trials, which could have occurred due to the fact that the same rewards were delivered twice on these trials during offer 1 and outcome epochs.

**Overlapping responses to offers and their predicted outcomes.** If OFC indeed reactivates outcome responses to encode offers during prospective evaluation[10,14], then we should expect overlapped neural response patterns to offers and outcomes. To compare response patterns, we examined the relationship between two sets of regression coefficients: one for offer-period firing rate against the size of offer 1 and the other for outcome-period firing rate against outcome size. We observed a positive correlation between these two sets of coefficients in both described ($r = 0.27$; $P = 0.003$; $n = 125$; Spearman's correlation; Fig. 4a,b) and experienced trials ($r = 0.36$; $P < 0.001$; $n = 125$; Spearman's correlation; Fig. 4c,d). We chose Spearman's correlation (instead of Pearson) to minimize the influence of the regression coefficients' unknown distribution and potential outliers. We also confirmed that none of the data points qualify as outlier with a Cook's D test (Supplementary Fig. 3). We confirmed the observation of a positive overlap in regression coefficients by implementing a permutation test (Fig. 4b,d, and Methods), and by using a multiple regression model that included the additional factor of choice for outcome epoch, which was also confirmed with permutation tests (Supplementary Fig. 4). Importantly, the strengths of reactivation responses, as measured by the Spearman's correlation coefficients, were not statistically different between described and experienced trials ($z$-value $= -1.10$; $P = 0.269$; $n = 2$; Fisher's Transformation Test). This result argues against the possibility that the described offer (a secondary reward, that is, coloured rectangle) elicits a weaker neural response than the experienced offer (a primary reward, that is, water aliquot). This result also argues against the possibility that the overlapped response between offer and outcome were due to the potentially common but weaker mouth movement during described offer epoch.

We then tested whether there is an overlap in the set of neurons involved in encoding offer 1 and in encoding outcome. To do so, we used a technique we devised and used for this purpose in earlier studies[20,21]. Specifically, we took the absolute value of the two sets of linear regression coefficients (mentioned above) as an index of task participation (that is, a measure of unsigned coding strength). If the same—or at least a positively overlapping—group of neurons participates in representing the values of offer and outcome, then the absolute value of the regression coefficients for offer and outcome will be positively correlated. Conversely, if there are distinct populations, we will observe a significant negative correlation between these variables. The reason lies in the fact that if there are separable populations,

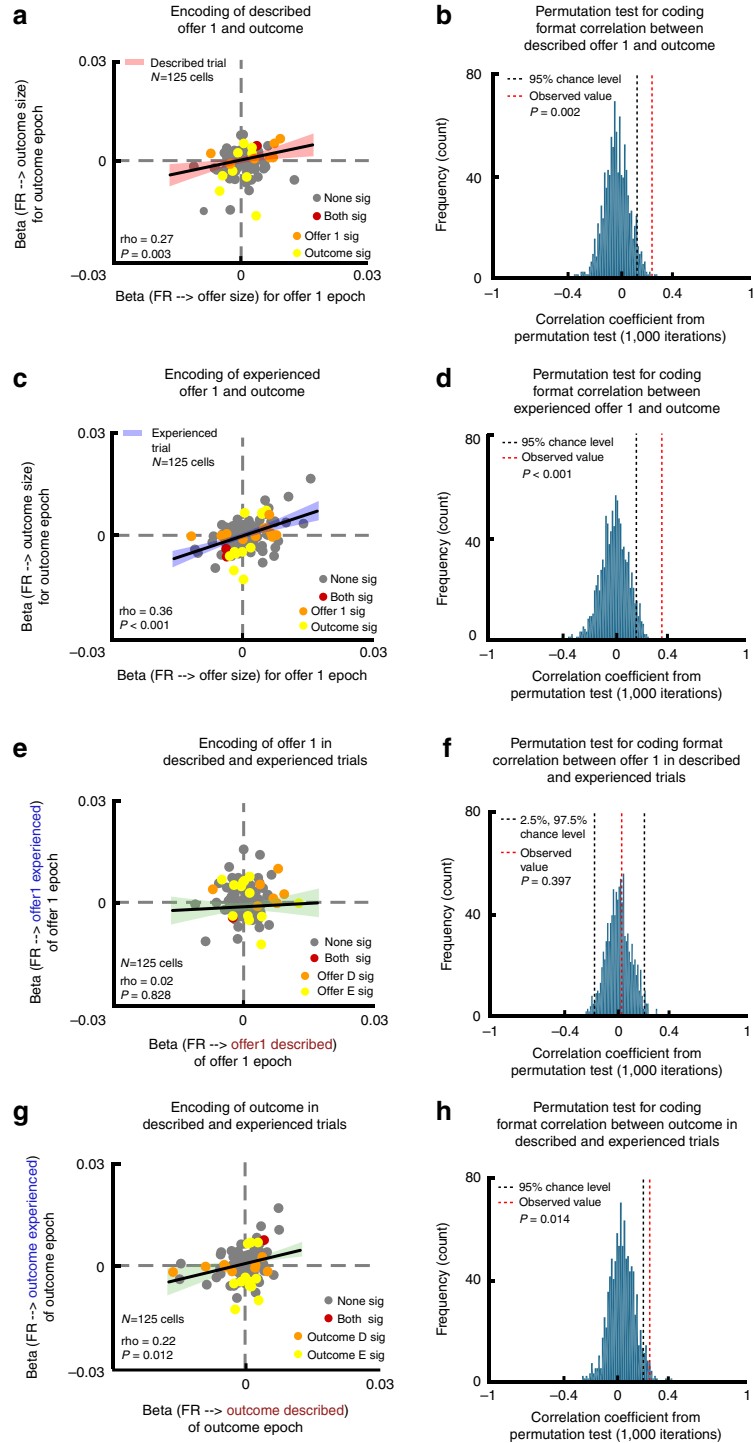

**Figure 4 | Reactivation Responses.** Scatter plots illustrate the correlation analyses used to assess reactivation of outcome neural response pattern during offer 1 encoding. (**a**,**c**,**e**,**g**) Each dot represents a neuron. Orange: the neuron is significantly tuned in the regression described on the *x*-axis. Yellow: the neuron is significantly tuned in the regression described on the *y*-axis. Red: the neuron is significantly tuned in the regressions described on both the *x*-axis and the *y*-axis. Grey: the neuron is not significantly tuned in either the regression described on the *x*-axis or the *y*-axis. (**b**,**d**,**f**,**h**) Permutation test of significance for the correlation coefficient between two sets of regression coefficients. (**a**,**b**) Reactivation for described offers: correlation between regression coefficients for offer size in offer 1 epoch (*x*-axis) and outcome size in outcome epoch (*y*-axis). Positive correlation indicates overlapped neural response pattern. (**c**,**d**) Reactivation for experienced offers: correlation between regression coefficients for offer size in offer 1 epoch (*x*-axis) and outcome size in outcome epoch (*y*-axis). Positive correlation indicates matching neural response pattern. (**e**,**f**) Unrelated coding for described and experienced offers: correlation between regression coefficients for described offer in offer 1 epoch (*x*-axis) and experienced offer in offer 1 epoch (*y*-axis). This lack of correlation is consistent with no overlap in coding scheme for described and experienced offers. (**g**,**h**) Similar coding for outcomes across described and experienced trials: correlation between regression coefficients for described-trial outcome in outcome epoch (*x*-axis) and experienced-trial outcome in outcome epoch (*y*-axis). This significant correlation indicates a positive overlap in coding scheme for described and experienced outcomes.

then stronger selectivity for one option implies weaker selectivity for the other one, and will therefore produce a negative correlation. Finally, if there is no special relationship between the populations, and parameter sensitivity is distributed randomly across the population, we will see no correlation between these variables. This analysis revealed a positive correlation between the unsigned regression coefficients for described ($r = 0.33$; $P < 0.001$; $n = 125$; Spearman's correlation) and experienced ($r = 0.19$; $P = 0.037$; $n = 125$; Spearman's correlation) trials. These results argue against the hypothesis that offers and outcomes are encoded by specialized sets of neurons; rather they suggest that a single set of neurons encodes both values at different times in the trial.

We next used a non-linear neural network decoding approach to confirm these findings. First, we defined a 125-dimensional neuronal space, with each neuron taking up one dimension. Second, we separated trials into 10 groups each corresponding to one of the five offer 1/outcome sizes in each trial type (described and experienced). Third, we computed the activation states for offer 1 and outcome epochs separately by randomly sampling one trial per neuron from each group and averaging the firing rates across time bins in each epoch. Finally, we trained the decoders on activation states associated with offer 1 and outcome epochs separately (see Methods).

For described trials, we found that a decoder trained on outcome responses could decode activation states of offer 1 at levels greater than chance (performance: 24.52%; $\chi^2 = 3.43$; $P = 0.03$; $n = 625$; effect size $= 1.30$; one-sided chi-square test; chance level: 20%, Fig. 5a). Equivalently, a decoder trained on population activity states for offer 1 could decode population activation states during outcome delivery (26.20%; $\chi^2 = 6.42$; $P = 0.006$; $n = 625$; effect size $= 1.42$; one-sided chi-square test; Fig. 5a). Similarly, for experienced trails, a decoder trained on population activation states during outcome epoch could decode activity patterns of experienced offer 1 (27.32%; $\chi^2 = 8.87$; $P = 0.001$; $n = 625$; effect size $= 1.51$; one-sided chi-square test; Fig. 5a). Equivalently, decoder trained on population activation states for offer 1 could decode neural activity patterns during outcome delivery (39.60%; $\chi^2 = 56.45$; $P < 0.001$; $n = 625$; effect size $= 2.63$; one-sided chi-square test, Fig. 5a). We showed in Supplementary Fig. 5a,b that the relatively low decoding accuracy was primarily caused by responses to smaller-sized offers, because subjects seldom chose and received those offers. We also tested these decoders with a sliding window of neural activation patterns from offer 1 epoch, demonstrating the temporal dynamics of the reactivation response (Supplementary Fig. 5c,d). Reactivation response occurred slightly later in described than in experienced trials.

To exclude the possibility that our results could be due to the particular decoding technique we chose, we also confirmed these results with a Support Vector Machine (SVM) decoder (see Methods). The SVM decoder was trained to distinguish, within each trial type, between the population activation state associated with each size of the outcome against those associated with the rest of other sizes of outcome, and then, tested on neural response patterns of offer 1, and *vice versa*, (Fig. 5d). After correcting for error rate, we found that a decoder trained on neural activation to outcomes in described trials could decode neural response for described offers (24.00%; $\chi^2 = 2.69$; $P = 0.05$; $n = 625$; effect size $= 1.26$; one-sided chi-square test); the same was observed in experienced trials (28.52%; $\chi^2 = 11.89$; $P < 0.001$; $n = 625$; effect size $= 1.95$; one-sided chi-square test). Similarly, a SVM decoder trained on neural response for described offer 1 could decode that for outcome delivery in described trials (28.44%; $\chi^2 = 11.67$; $P < 0.001$; $n = 625$; effect size $= 1.95$; one-sided chi-square test); the same was observed in experienced trials (43.84%; $\chi^2 = 80.64$; $P < 0.001$; $n = 625$; effect size $= 3.12$ one-sided chi-square test).

**Overlapping response to outcomes across trial types**. We next examined how neural responses to outcomes on the two trial types related to each other. We predicted that neural activations to outcomes multiplex the associative structure specific and the reward general response patterns. Therefore we predict some overlap in the neural activations to outcomes, even though they come from distinct offer types. Supporting the idea of an overlap, we observed positively correlated tuning patterns for outcomes on described and experienced trials ($r = 0.22$; $P = 0.012$; $n = 125$; Spearman's correlation; Fig. 4g,h). We also found an overlapping subset of OFC neurons encoding the outcomes on the two trial types, indicating a lack of neuronal specialization for the two groups of outcome ($r = 0.21$, $P = 0.020$; $n = 125$; Spearman's correlation). Supporting the reactivation hypothesis, we found that a decoder trained on neural activation to outcomes in described trials could decode neural responses to outcome in experienced trials better than chance (31.96%; $\chi^2 = 22.63$; $P < 0.001$; $n = 625$; effect size $= 1.88$; chi-square test; Fig. 5c; chance level: 20%). A decoder trained on neural activation states for outcome in experienced trials, however, could not significantly decode activation states for outcome in described trials (22.04%; $\chi^2 = 0.67$; $P = 0.21$; $n = 625$; effect size $= 1.13$; chi-square test; Fig. 5c). We suspect that high noise in training data contributed to this asymmetry in decoding (Supplementary Note 1). Thus, together, these results indicate some overlap in coding schemes for outcomes in described and experienced trials types.

**Non-overlapping responses to offers across trial types**. We have shown above that OFC reactivates neural response to outcomes during prospective evaluation. However, whether the reactivated neural response was reward general or unique to each specific associative structure remained unaddressed. We hypothesized that during prospective evaluation, only the associative structure specific response is represented. Since the size of described and experienced offer 1s was revealed through different offer-outcome associative event sequences, we would expect no correlation between the neural responses they elicit, even if they predicted the same reward.

As above, to compare tuning patterns, we computed the regression coefficients for normalized firing rate against the size of offers separately in the described and experienced conditions. We observed no correlation between the two sets of regression coefficients ($r = 0.02$; $P = 0.828$; $n = 125$; Spearman's correlation; Fig. 4e,f). Moreover, in comparison, correlation coefficient between regression coefficients for described offers and experienced offers is significantly smaller than that between described and experienced outcomes ($z$-value $= 2.25$; $P = 0.012$; $n = 2$; Fisher's Transformation Test). The similar effect was observed in comparison to correlation coefficient between regression coefficients for described offers and outcomes ($z$-value $= 2.84$; $P = 0.002$; $n = 2$; Fisher's Transformation Test) and that between experienced offers and outcomes ($z$-value $= 3.94$; $P < 0.001$; $n = 2$; Fisher's Transformation Test). Thus, OFC recruited unrelated encoding patterns for offers that were presented with different associative event sequences, even if they predicted the same reward. This lack of correlation was not due to lack of power or spurious distribution of the coefficients. We performed a power analysis and a permutation test (Fig. 4f; see Methods for details) and both analyses suggested that given our sample size, if a significant correlation truly existed, we would have observed a

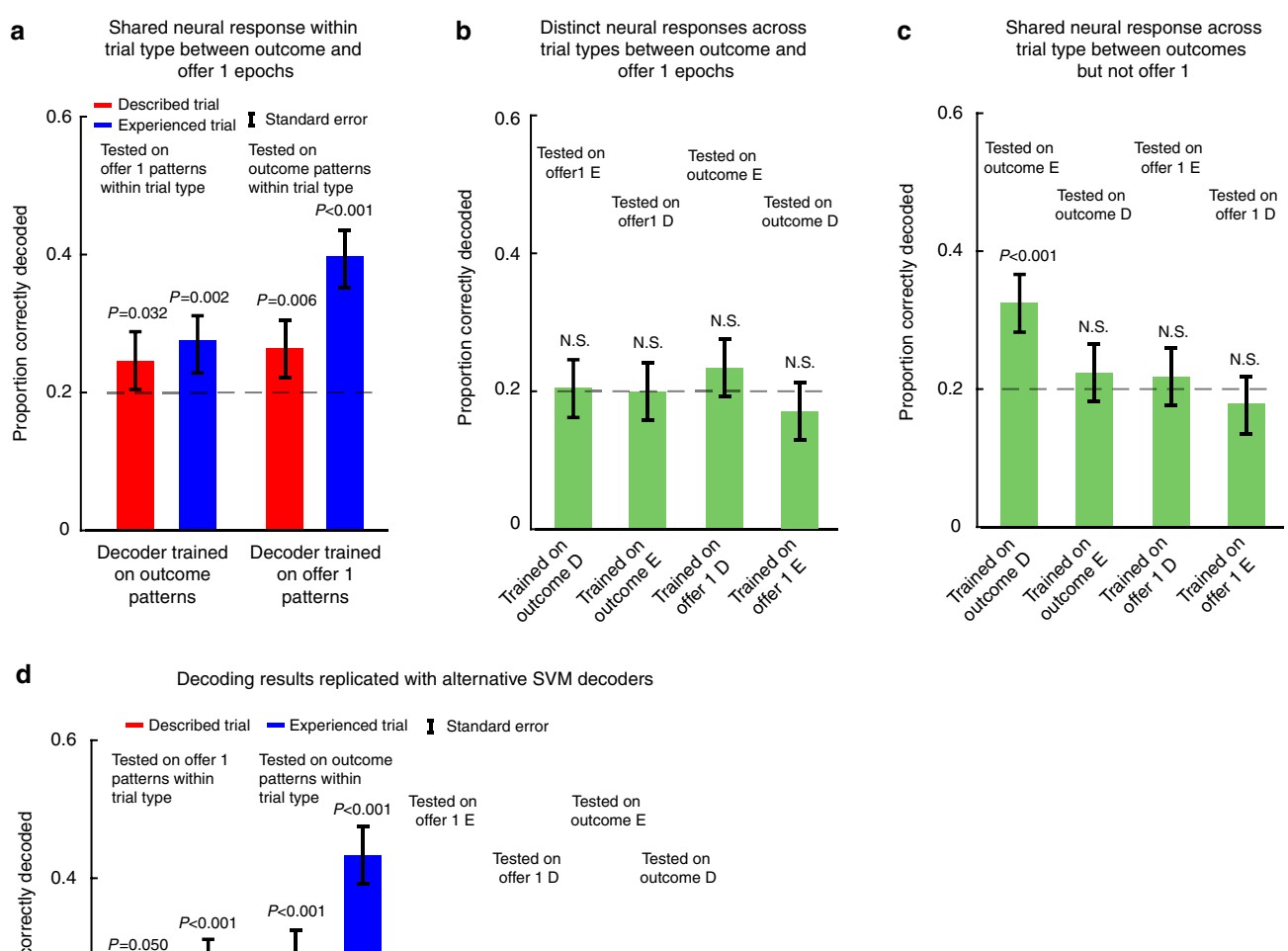

**Figure 5 | Decoding Accuracy.** D: described trials. E: experienced trials. (**a**) Decoding results indicate reactivation response within each trial type (described or experienced). The decoders trained on population activation states for outcome could successfully decode offer size from neural response patterns associated with offer 1 within each trial type, and vice versa when the decoders were trained on population activation states during offer 1. (**b**) No reactivation response is observed across trial type using decoding approach. (**c**) Neural response patterns for outcomes in described and experienced trials share coding scheme. Neural response patterns for described and experienced offers do not share coding scheme. (**d**) Decoding analyses using Support Vector Machine (SVM) replicate findings in (**a,b**) with neural network decoder.

correlation coefficient (effect size) between $-1$ to $-0.19$ and $0.19$ to $1$, instead of $0.02$.

We also observed no correlation between the unsigned regression coefficients ($r = 0.002$; $P = 0.98$; $n = 125$; Spearman's correlation). Therefore, selectivity for described and experienced offers recruited neurons randomly distributed across the population instead of a single subset.

The decoding approach showed similar results. Specifically, we found that a decoder trained on population activation states for described offer 1 could not decode population activation states for experienced offer 1 ($21.56\%$; $\chi^2 = 0.37$; $P = 0.27$; $n = 625$;

effect size $= 1.10$; chi-square test; Fig. 5c). Similarly, a decoder trained on population activation patterns for experienced offer 1 could not decode population activation patterns for described offer 1 ($17.36\%$; $\chi^2 = 1.27$; $P = 0.87$; $n = 625$; effect size $= 0.85$; chi-square test; Fig. 5c).

Furthermore, although we observed that outcome and offers within the same trial type showed significantly overlapping population activation states, we did not observe this overlap across trial types. Specifically, a decoder trained on responses to outcomes in described trials could not decode neural activations to experienced offers ($20.60\%$; $\chi^2 = 0.04$; $P = 0.42$; $n = 625$; effect

size = 1.04; one-sided chi-square test; chance level: 20%; Fig. 5b). Likewise, a decoder trained on neural activations to outcomes in experienced trials could not decode neural activations to described offers (18.40%; $\chi^2 = 0.42$; $P = 0.74$; $n = 625$; effect size = 0.90; one-sided chi-square test; Fig. 5b). Similarly, a decoder trained on neural activations to described offers could not decode neural activations to outcomes on experienced trials (23.24%; $\chi^2 = 1.75$; $P = 0.09$; $n = 625$; effect size = 1.21; one-sided chi-square test; Fig. 5b). And a decoder trained on neural activations to experienced offers could not decode neural activations to outcomes in described trials (17.12%; $\chi^2 = 1.53$; $P = 0.89$; $n = 625$; effect size = 0.83; one-sided chi-square test; Fig. 5b).

**Principal component trajectories of two trial types.** Our central hypothesis predicts that the OFC population neural responses should reflect different associative structures in described versus experienced trials during prospective evaluation, and this difference should reduce as trial proceeded to outcome delivery.

To test this prediction, we used a dimensionality reduction approach. We first defined a 125-dimensional neuronal space, with each neuron taking up one dimension. Then we computed the activation state for each of five 300 ms epochs (offer 1 cue, offer 1 value, offer 2, choice and outcome) in each trial type, by averaging firing rates for each neuron across all trials and across time bins in each epoch. We subsequently conducted a principal component analysis on the 125-dimensional, 5-epoch, 2-trial-type, population responses.

We found that the top three principal components could together account for 71.68% of the variance in the data (Fig. 6a). Next we plotted the trajectories of the population activation states as trial proceeded for described and experienced trials separately in the top-three-PC space. We also plotted the averaged trajectories from neural activation states with 1,000 iterations of permutated described versus experienced trial types. As shown in Fig. 6b, actual data showed mirrored but distinct activation state trajectories in described and experienced trials, with the distance between states being most prominent during prospective evaluation epochs, gradually reducing thereafter, and becoming most diminished after choice execution in outcome epoch. In contrast, the permutated described and experienced trajectories perfectly overlapped with each other. This result is in line with our prediction that the variances in population neural activities would reflect the distinct associative structures during prospective evaluation, potentially for guiding choice behaviour, and the differences gradually diminished as choice was carried out and the reward outcome delivered.

To formally test the change in distance between population activation states in described versus experienced trials as a function of trial progress, we re-defined population activation states for each trial type based on a sliding 300-ms bin from offer cue onset to the end of outcome delivery. Next we calculated and plotted the Euclidean distance between activation states from the two trial types (Fig. 6c). We then calculated the Euclidean distance from 1,000 sets of permutated data and plotted the mean and both the top and the bottom 2.5% significance cutoffs (Fig. 6c). We confirmed that the distance between activation states from two trial types were significantly larger than expected by chance during prospective evaluation epochs (for example, as in Fig. 6c, offer 1 cue epoch at 0.7 s: Euclidean Distance = 4.98, $P < 0.001$; offer 1 value epoch at 1.2 s: Euclidean Distance = 3.92, $P < 0.001$; offer 2 epoch at 2 s: Euclidean Distance = 3.05, $P < 0.001$) and then the distance reduced to below significance after choice and during outcome delivery (for example, as in Fig. 6c, choice epoch at 3 s: Euclidean Distance = 2.73, $P = 0.082$; outcome epoch at 4 s: Euclidean Distance = 1.80, $P = 0.679$).

## Discussion

We examined the relationship between ensemble neural responses to offers and outcomes in Area 13 of OFC in macaques, while they completed a riskless choice task. Our task used two trial types: described offer and experienced offer trials. Within each trial type, we found an overlap in coding scheme (meaning similar tuning strength and direction), for each offer and its corresponding outcome. We also found an overlap between the two outcome responses across trial types, indicating that OFC carries a general reward signal. However, we observed unrelated coding schemes for the responses to the two types of offers. These three patterns are consistent with our hypothesis that OFC reactivates neural responses to outcomes that are specific to associative structures during prospective evaluation, but it encodes the delivered reward outcome after a choice with both an associative structure-specific and a reward-general signal that is conserved across outcomes with distinct preceding associative event sequences.

Our theory offers a potential reconciliation for two different and seemingly inconsistent sets of results. On one hand, it appears that representation of reward-predicting stimuli reactivates similar neural response pattern as the primary reward does[10,11,14,15,22,23]. On the other hand, it appears that OFC calls upon associative structure specific neural responses during prospective evaluation to direct behaviour[16–18]. Our findings suggest that responses to offers involve a partial reactivation of the responses to outcomes; the reactivated part is specific to the offer-outcome associative event sequence. Responses to outcomes multiplex the associative structure-specific signal with a more general reward coding that is the same regardless of the associative structure that predicted it.

Thus, when different associative structures are used to present offers that predict the same reward, OFC recruits unrelated coding schemes for each during offer presentation[17]. However, when different visual stimuli but the same associative event sequence is used to predict the same reward, the associative structure specific value encoding during stimuli presentation should resemble a reactivation[10,22]. Relatedly, outcome specific and outcome general reward representation are double dissociable, both behaviourally and neurally[24,25]. For example, OFC lesions specifically impair outcome specific reward value representation and abolish its effect on later blocking and devaluation tests[8]. Although our task could not directly test this aspect, a generalization from our results suggests that outcome-specific reward representation would be represented as part of the associative structure specific representation during both offer and reward outcome epochs. (For more discussion on these subjects, see[8,12,26,27].)

Our results are consistent with the cognitive map theory of OFC functions, which states that OFC instantiates a cognitive map of the task space, meaning that it represents, on the fly, the associative structures that are relevant to solving the current task[5–7]. Lesion studies show that OFC is necessary for using knowledge about the associative structure to guide goal-directed behaviour in both decision-making and learning[3,5–7,9]. However, less is known about how OFC represents associative structures and how this representation is involved in guiding goal-directed behaviours such as reward-based decision-making. A recent fMRI study showed that different hidden task states, or underlying associative structures, can be decoded from human mOFC and the decoding accuracy was positively correlated with behavioural performance in the task[28]. Our finding, that OFC encodes the reward in associative structure specific format during prospective evaluation, suggests that OFC emphasizes how the reward-predicting events will unfold and how to obtain the reward, or in reinforcement learning terms, it represents accurate state and

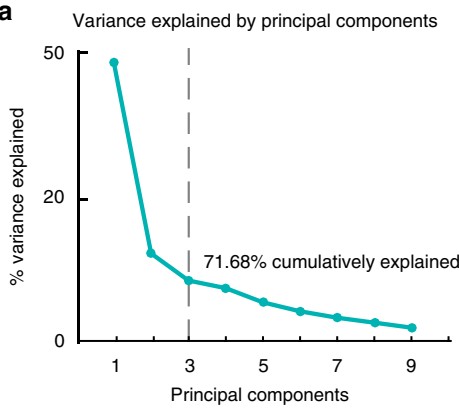

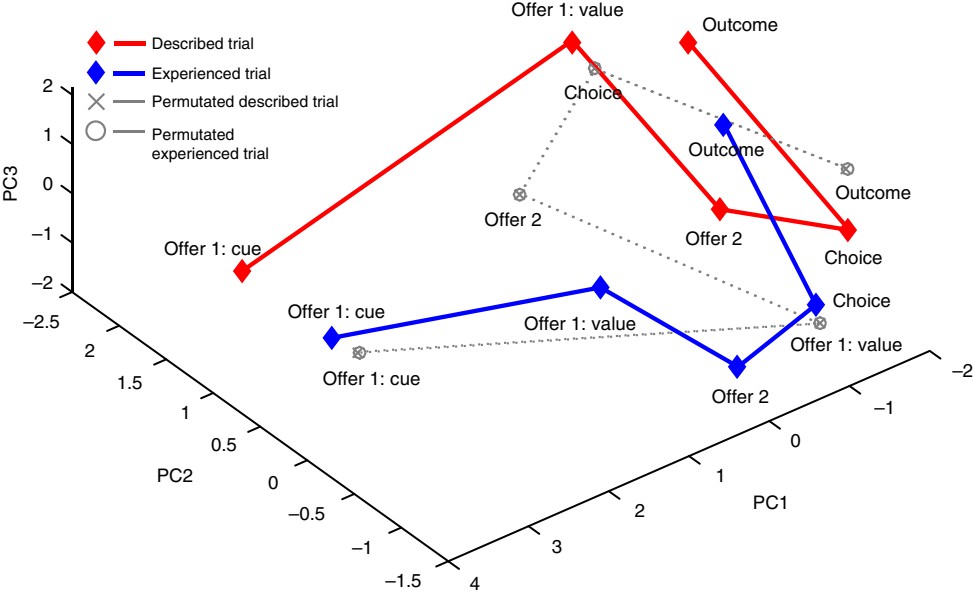

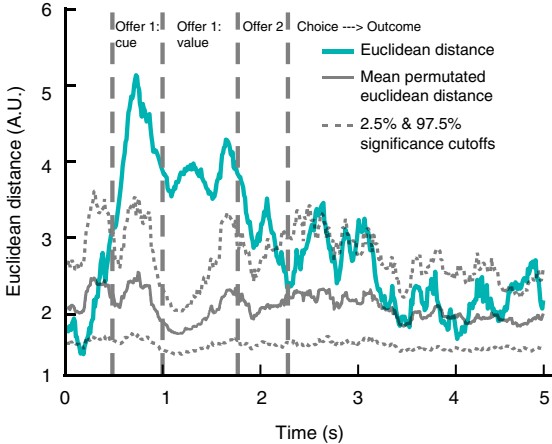

**Figure 6 | Principal Component Trajectories.** Population activation states from described and experienced trials converged as trial proceeded. (**a**) Scree plot: total variance in the data explained by number of principal components. Top three principal components together explained 71.68% of the variance in the data. (**b**) Trajectories based on the top three principal components: population activation states from the two trial types took up mirrored but separate coordinates in the top PC space, which gradually converged as trial proceeded. (**c**) Euclidean distance between population activation states from described and experienced trials: data are aligned at the beginning of cue epoch for offer 1 (x-axis). Population activation states were most separated during prospective evaluation; this separation reduced to below significance when choice was carried out and reward was delivered. Dotted grey lines indicate significance cutoffs calculated from 1,000 iterations of permutation test and the solid grey line indicates mean Euclidean distance from permuted data.

reward expectations to guide action selection during prospective evaluation. Subsequently, OFC uses both associative structure specific and reward general encodings during post-choice phase (reward outcome delivery), suggesting that this multiplexed learning signal is potentially used to update or reinforce the current associative structure during reward delivery.

It is important to note that, although OFC lesion in rodents impairs performance in a broad set of tasks that rely on cognitive map representation, such as reinforcer devaluation, reversal learning, and Pavlovian-instrumental transfer[7,9], the results in monkeys are more heterogeneous. For example, excitotoxic lesion of medial OFC in monkeys impaired performance only in reinforcer devaluation but not reversal learning[29]. One possibility is that reversal learning relies on the adjacent lateral OFC in monkeys[30,31]. Therefore, it is hard to tell whether our results will generalize to other sub-regions of OFC. Speculatively, these various sub-regions of OFC in monkeys may support representations of different aspects of the associative structure or the cognitive map. This possibility calls for direct test in future research.

Relatedly, recent studies have greatly enriched our understanding of OFC function. OFC is now considered as a crucial region to a broad spectrum of goal-directed behaviours[9,28,32–40]. Moreover, the involvement of OFC in such a variety of goal-directed behaviours suggests that OFC may be part of a broader frontal network underlying goal-directed learning and decision-making, including economic choice[41–43], rather than being a pure value region[44,45]. Consistent with these views, our results suggest that OFC (at least Area 13) recruits the associative structure specific neural activations to encode offers prospectively to guide subsequent choice behaviour. An intriguing venue for future research would be investigating OFC's role in goal-directed behaviour as a part of the proposed distributed network[43].

The behavioural data are interesting by themselves. We observed that monkeys are more accurate at choosing the larger reward on experienced trials than on described trials. This result is consistent with previous findings showing that gambles whose statistics are based on description and on experiences are processed in different ways in humans[46,47] and monkeys[48]. This observation might also reflect higher uncertainty in the reward representation for described offers where the dynamic pairing of offer cue and one of the five values was not directly observable but inferred, whereas the pairing in experienced trials was directly observable. Alternatively, the modality of the offer may affect the way it is framed: the way in which an offer is presented—or framed—can measurably affect preferences in humans[49] and monkeys[50–54]. Future research will be required to disambiguate these possibilities.

## Methods

**Subjects.** Two male rhesus macaques (Macaca mulatta) served as subjects to the current experiment. All animal procedures were approved by the University Committee on Animal Resources at the University of Rochester and were designed and conducted in compliance with the Public Health Service's Guide for the Care and Use of Animals.

**Recording site.** A Cilux recording chamber (Crist Instruments) was placed over the area 13 (ref. 55) of OFC (Fig. 1b and Supplementary Fig. 1). The targeted area expands along the coronal planes situated between 28.65 and 33.60 mm rostral to the interaural plane with varying depth. Position was verified by magnetic resonance imaging with the aid of a Brainsight system (Rogue Research Inc.). Neuroimaging was performed at the Rochester Center for Brain Imaging, on a Siemens 3T MAGNETOM Trio Tim using 0.5 mm voxels. We confirmed recording locations by listening for characteristic sounds of white and grey matter during recording, which in all cases matched the loci indicated by the Brainsight system.

**Electrophysiological techniques.** Single electrodes (Frederick Haer & Co., impedance range 0.8–4 MU) were lowered using a microdrive (NAN Instruments)

until waveforms of between one and five neuron(s) were isolated. Individual action potentials were isolated on a Plexon system (Plexon). We defined a priori our sample size of the current study with a power analysis. Specifically, power analysis estimates the minimum sample size required to detect an effect of a given size with a certain degree of confidence (significance level, that is, probability of Type I error, and, power, that is, 1 minus probability of Type II error). To estimate the effect size, we used the mean effect size of a previous study from our lab that recorded in the same region (Area 13 of OFC) and conducted the same ensemble analysis as in the current study[20]. In this previous study, mean effect size of significant correlations between two sets of regression coefficients is $r = 0.386$ (effect size of all significant correlations reported in the paper: 0.68, 0.33, 0.41, 0.31 and 0.2). We used 0.05 as significance level and 0.85 as power. A power analysis with these parameters suggests that the minimum sample size required to detect an effect size of 0.386 with significance level 0.05 and power 0.85 is $n = 57$. To replicate the same effect in two animals, our goal was to collect at least 57 neurons from each animal. Eventually, we collected 65 and 60 neurons from each animal, respectively.

Neurons were selected for study solely based on the quality of isolation; we never preselected based on task-related response properties. All collected neurons for which we managed to obtain at least 399 trials were analysed; no neurons were excluded from analysis.

**Eye tracking and reward delivery.** Eye position was sampled at 1,000 Hz by an infrared eye-monitoring camera system (SR Research). Stimuli were controlled by a computer running MATLAB (Mathworks) with Psychtoolbox[56] and Eyelink Toolbox[57]. A standard solenoid valve controlled the duration of juice delivery. The relationship between solenoid open time and juice volume was established and confirmed before, during, and after recording.

**The riskless choice task.** Each trial started with an initial eye fixation on a white dot (radius: 10 pixels) at the center of the screen (Fig. 1a, resolution, 1,024 × 768). After 200 ms, the offer 1 cue appeared on the screen (rectangle 300 × 80 pixels, 11.35 × 4.08 DVA) for 500 ms. A grey cue indicated that the forthcoming offer 1 would be in a described format; a white cue indicates that the offer 1 would be in an experienced format.

On described trials, offer 1 size was revealed via the presentation of a rectangle with one of the five colours (red, yellow, blue, green, cyan) during offer 1 epoch; each colour predicted an reward size (75, 100, 150, 200, 250 µl water reward). On experienced trials, the screen remained blank and subjects received an aliquot of water equal to the offered size and thus gained information about the offer size directly. The set of possible offer 1 sizes were matched for the two trial types. The offer 1 epoch lasted for 750 ms.

Subsequently, offer 2 appeared. Offer 2 came in three sizes (150, 175, 200 µl water reward); the size was indicated by a natural scene picture appearing on the opposite side of the screen from the offer 1 (rectangle 300 × 80 pixels, 11.35 × 4.08 DVA). The offer 2 epoch lasted for 500 ms.

After another 200 ms fixation, both options, the offer 1 cue (a grey rectangle on described trials and a white rectangle on experienced trials) and offer 2 (the natural scene picture), reappeared in their original positions. Thus, subjects need to maintain the value of offer 1 in working memory to choose successfully. The subject chose an option by fixating on it for 300 ms. A magenta frame then appeared around the chosen option (300 ms). The chosen reward was then delivered at the beginning of the 750 ms outcome epoch started. A 1,000-ms black-screen inter-trial interval followed. The trial type (experienced or described), offer position, offer 1 size and offer 2 size were all randomized independently for each trial.

We defined associative structures in this task as the modalities and associative event sequences with which offer 1 size was revealed. Specifically, for described offer 1, its size was revealed via a visual cue, in a stimulus-stimulus association (that is, a grey rectangle followed by one of the five coloured rectangles, forming a stimulus to conditioned reinforcer/secondary reward associative event sequence). For experienced offer 1, its size was revealed via a gustatory cue (a primary reward), in a stimulus-reward association (that is, a white rectangle followed by one of the five sizes of water reward, forming a stimulus to primary reward associative event sequence).

No blinding procedure was done.

**Statistical methods.** All choices were counted as correct when subjects selected an option with value greater than or equal to the non-chosen alternative. Chance level of correct choice rate (56.67%) was calculated based on experimental design and each possible combination of offer 1 and 2 sizes. Chi-square test, binomial test, and power analysis were conducted using R. Log odds, relative risk, $R^2$, and Hedge's G were reported as the effect size for chi-square test, binomial, linear regression, and t-test, respectively. Subjects' choice behaviour was fitted using a logistic regression model and was conducted using MATLAB (Mathworks).

PSTHs were constructed by aligning spike rasters to the presentation of the offer 1. Firing rates were calculated in 10 ms bins but were generally analysed in longer epochs. For display, PSTHs were smoothed using a 200 ms running boxcar.

For all regression analyses fitting firing rates against predictor of interest, the firing rates were normalized (z-scored) for each neuron to avoid spurious correlations. The proportion of neurons tuned for each predictor of interest (described offer size, experienced offer size and outcome size) was determined

based on linear regression analysis, fitting normalized firing rates from the event-related epoch against each single predictor of interest:

$$FR_{norm} = B_1 \cdot Predictor + intercept.$$

To test for reactivation response, we first selected trials in which offer 1 was chosen. Based on the selected trials, we fitted the following linear regression models with normalized firing rates from event-related epochs:

$$FR_{norm} = B_{OFR.D} \cdot described\ offer\ size + intercept;$$

$$FR_{norm} = B_{OFR.E} \cdot experienced\ offer\ size + intercept;$$

$$FR_{norm} = B_{OTC.D} \cdot described\ outcome\ size + intercept;$$

$$FR_{norm} = B_{OTC.E} \cdot experienced\ outcome\ size + intercept.$$

These regression coefficients from the entire sample contain information about population tuning formats (strength and direction). Therefore, we used Spearman's correlation between $B_{OFR.D}$ and $B_{OTC.D}$ for described trials, and between $B_{OFR.E}$ and $B_{OYC.E}$ for experienced trials, to measure the similarity in coding format and thus reactivation of outcome responses during offer 1 epoch. We chose Spearman's correlation (instead of Pearson) to minimize the influence of the regression coefficients' unknown distribution and potential outliers.

Subsequently, we compared the neuronal participation in signalling offers and outcomes by correlating absolute value of $B_{OFR.D}$ and absolute value of $B_{OTC.D}$ for described trials, and then, absolute value of $B_{OFR.E}$ and absolute value of $B_{OTC.E}$ for experienced trials.

Finally, we also compared encoding patterns and neuronal involvement for signalling two offers and two outcomes by correlating the signed and absolute values of $B_{OFR.D}$ and $B_{OFR.E}$, and then, $B_{OTC.D}$ and $B_{OTC.E}$.

As the correlation analysis was performed on regression coefficients whose distribution was unknown, we also tested the significance of the observed correlation coefficients using a permutation test. For the permutation test, all regression was re-conducted by keeping the normalized firing rates the same as in the original analysis but randomizing the predictors in each of the regression model above. Then we correlated the permutation regression coefficients. Subsequently, we compared the correlation we observed against those from 1,000 iterations of the permutation test. The significance cutoff was set as higher than 95% of the correlation coefficients from the permutation analysis.

To test for reactivation response using alternative regression models, we included all trials for analysis, instead of selecting only offer1-chosen trials. We then fitted the following linear regression models with normalized firing rates from event-related epochs:

$$FR_{norm} = B_{OFR.D} \cdot described\ offer\ size + intercept;$$

$$FR_{norm} = B_{OFR.E} \cdot experienced\ offer\ size + intercept;$$

$$FR_{norm} = B_{OTC.D} \cdot described\ outcome\ size + B_2 \cdot choice + intercept;$$

$$FR_{norm} = B_{OTC.E} \cdot experienced\ outcome\ size + B_2 \cdot choice + intercept.$$

Choice was defined as a binary variable of choosing either offer 1 or 2. The first two of this set of regression models included only offer size as a single predictor, since no other meaningful predictors had been revealed yet during offer 1 presentation. The remaining regression models included both outcome size and choice as predictors, since choice is a prominent confounding predictor besides outcome size during outcome epoch, and this regression model allows us to test the encoding for outcome size while controlling for choice. Subsequently, we compare the encoding patterns for offers and outcomes by correlating $B_{OFR.D}$ and $B_{OTC.D}$ for described context, and then, $B_{OFR.E}$ and $B_{OTC.E}$ for experienced context. Since the correlation analysis was on regression coefficients whose distribution was unknown, we also tested the significance of the observed correlation coefficients using a permutation test.

Fisher's transformation test was used to compare two correlation coefficients. For paired sample, $z$-value is calculated according to:

$$r'_1 = 0.5 \log\frac{1+r_1}{1-r_1}; \quad r'_2 = 0.5 \log\frac{1+r_2}{1-r_2};$$

$$z = \frac{r'_1 - r'_2}{1/\sqrt{n-3}};$$

where $n$ is the sample size.

**Decoding analyses.** For the decoding analysis, we chose a non-linear neural network decoding technique that is considered to perform well in non-linear, multiclass classifications[58–62]. We chose the non-linear decoder because the population neural response in frontal cortex is considered to be highly multiplexed and non-linear, and, the classification of neural activity on offer sizes in the current data set is multi-way (five offer sizes) instead of binary. We also replicated the decoding results with a more standard SVM as error-correcting output codes multiclass model (https://www.mathworks.com/help/stats/classificationecoc-class.html).

To generate population activation states for the decoding analysis, we first separated all trials of each neuron by offer size (5) × trial type (2) and therefore

into 10 groups. On average, we obtained 45 trials in each group. We then randomly sampled one trial out of each group. Subsequently, we averaged normalized firing rates from the selected trial for each event-related epoch (offer 1 and outcome) and for each neuron. We then polled all 125 neurons' averaged response during each epoch to generate one population activation state for that particular epoch. We sampled one trial with replacement from each group for each neuron independently and generated in total 500 population activation patterns for offer 1 and outcome epochs. The number 500 was chosen because neural network decoder is computationally expensive and its training requires relatively large set of exemplars[61,62].

We separated the population activation states into training and testing subsets following a four-fold cross-validation procedure, leading to four sets of 375 training population activation states and 125 testing population activation states. Note that even though independent sampling with replacement for each neuron might lead to small overlap in population activity patterns between training and testing sets, all test sets were only used to determine that our decoders were successfully trained to reach high performance and were never used to test for main hypothesis. All of our main analyses involved training the decoder with neural response from outcome epoch and then testing with neural response from offer epoch, and vice versa. Due to the fact that subjects rarely chose and received smaller-sized offers during outcome epoch, population activation states for smaller-sized outcomes include only response from neurons with corresponding data.

For the non-linear neural network decoding analyses, there are three layers in the network: an input layer with 125 nodes taking in one population activation pattern; a hidden layer with 40 nodes connected to the input layer and the output layer; an output layer with 5 nodes each corresponding to one of the five sizes of offer 1/outcome. The non-linear neural network decoders were trained with standard back-propagation algorithm[62,63]. The neural networks' weights were initialized as a small random number between − 0.01 and + 0.01. Total number of training epochs was 1,000. A single run through the back-propagation algorithm contains one forward pass and one backward pass.

During the forward pass, the activation of each layer was calculated as the weighted sum of the previous layer with a transformation activation function. The activation of the whole input layer is one population activation state. Activation of each node corresponds to response of one neuron:

$$x_i(t) = population\ activation\ pattern_i(t);$$

$x_i(t)$ is the activation of the $ith$ input units which equals to the neural response of the $ith$ neuron in the $tth$ population activation state.

The activation of hidden layer is the weighted sum of input layer transformed with a logistic activation function:

$$s_j(t) = \sum_1^i w_{ji}(n)x_i(t);$$

$$h_j(t) = \frac{1}{1+e^{-s_j(t)}} = \frac{1}{1+e^{-\sum_1^i w_{ji}(n)x_i(t)}};$$

$s_j(t)$ is the hidden unit j's weighted sum input from the input layer. $h_j(t)$ is the activation of the $jth$ hidden unit, which is $s_j(t)$ transformed with a logistic activation function. $w_{ji}(n)$ is the weight on the connection between input unit $i$ and hidden unit $j$ during the $nth$ training epoch.

The activation of the output layer is the weighted sum of hidden layer transformed with a softmax activation function:

$$s_k(t) = \sum_1^j w_{kj}(n)h_j(t);$$

$$y_k(t) = \frac{e^{s_k(t)}}{\sum_1^p e^{s_p(t)}} = \frac{e^{\sum_1^j w_{kj}(n)h_j(t)}}{\sum_1^p e^{\sum_1^j w_{pj}(n)h_j(t)}};$$

$s_k(t)$ is the output unit k's weighted sum input from the hidden layer. $w_{kj}(n)$ is weight on the connection between hidden unit $j$ and output unit $k$ during the $nth$ training epoch. $y_k(t)$ is the activation of the $kth$ output unit, which is $s_k(t)$ transformed with a softmax activation function based on activation of all of the $p$ output units (here $p = 5$).

During the backward pass, partial derivatives were calculated to update the weights between the output layer and the hidden layer and the weights between the hidden layer and the input layer. In a generic form, weight update uses gradient ascent on the log likelihood function:

$$w_{ab}(n+1) = w_{ab}(n) + \varepsilon\frac{\partial\log L(n)}{\partial w_{ab}(n)};$$

$w_{ab}(n)$ is weight on the connection between unit $b$ in the layer preceding the weights and unit $a$ in the layer succeeding the weights during the $nth$ training epoch. $\varepsilon$ is the learning rate that equals to 0.005.

In multi-way classification with softmax, the class given the input $x(t)$ has a multinomial distribution:

$$p(y^*(t)|x(t)) = \prod_{c=1}^{C} y_c(t)^{y_c^*(t)};$$

where $c$ indexes the classes and $C$ is the number of possible classes. $y^*$ is the target output or correct class label. The log likelihood function of this multinomial distribution is:

$$\log L = \sum_t \sum_c y_c^*(t) \times \log y_c(t).$$

To update weights between the output units and the hidden units:

$$\frac{\partial \log L(n)}{\partial w_{kj}(n)} = \sum_t \frac{\partial \log L(t)}{\partial s_k(t)} \times \frac{\partial s_k(t)}{\partial w_{kj}(n)};$$

where

$$\frac{\partial \log L(t)}{\partial s_k(t)} = \sum_p \frac{\partial \log L(t)}{\partial y_p(t)} \times \frac{\partial y_p(t)}{\partial s_k(t)}$$

$$= \sum_p \frac{y_p^*(t)}{y_p(t)} \times y_p(t) \times (\delta_{kp} - y_k(t));$$

$\delta_{kp} = 1$, if $p = k$; otherwise, it equals 0. Again, $y_p^*(t)$ is the value for the correct class label for the $p$th output unit corresponding to the $t$th population activation state. $y_p(t)$ is the actual neural network output value for the $p$th output unit. And

$$\frac{\partial s_k(t)}{\partial w_{kj}(n)} = h_j(t).$$

To update weights between the hidden units and the input units:

$$\frac{\partial \log L(n)}{\partial w_{ji}(n)} = \sum_t \frac{\partial \log L(t)}{\partial h_j(t)} \times \frac{\partial h_j(t)}{\partial s_j(t)} \times \frac{\partial s_j(t)}{\partial w_{ji}(n)};$$

where

$$\frac{\partial \log L(t)}{\partial h_j(t)} = \sum_k [y_k^*(t) - y_k(t)] \times w_{kj}(n);$$

$W_{kj}(n)$ is the weight on the connection between hidden unit $j$ and output unit $k$ during the $n$th training epoch. $y_k^*(t)$ is the value for the correct class label for the $k$th output unit corresponding to the $t$th population activation state. $y_k(t)$ is the actual neural network output value for the $k$th output unit. And

$$\frac{\partial h_j(t)}{\partial s_j(t)} = h_j(t) \times [1 - h_j(t)];$$

$$\frac{\partial s_j(t)}{\partial w_{ji}(n)} = x_i(t).$$

As defined above, $h_j(t)$ is the activation of the $j$th hidden unit, and, $x_i(t)$ is the activation of the $i$th input units which equals to the neural response of the $i$th neuron in the $t$th population activation state.

In other words, the non-linear neural network decoder takes population activation patterns (a $125 \times 1$ vector) as input, computes through one hidden layer of 40 hidden units with the logistic activation function, and then classifies the activation of the hidden layer into one of five offer sizes with the softmax activation function at the five-unit output layer. Decoders were each trained on population activation states for either described or experienced trials. Final decoding accuracy was determined as the averaged accuracy of four cross-validation sets.

An additional set of decoding analyses was run using SVM. These analyses utilized the Statistics and Machine Learning Toolbox of MATLAB. In short, to perform multi-way classification, we trained the SVM decoders as error-correcting output codes multiclass model (http://www.mathworks.com/help/stats/fitcecoc.html), to classify each population activation state as representing one size of offer 1/outcome versus all other sizes of offer 1/outcome. The same population activation states generated above for non-linear neural network decoding were used to train SVM. All of our decoding analyses using SVM involved training the decoder with neural response in outcome epoch and then testing with neural response in offer epoch, and vice versa.

**Principal component analysis.** We first defined the population activation state as a 125-dimensional vector, with each neuron taking up one dimension. Then we computed the activation state for each of five 300-ms epochs (offer cue, offer 1, offer 2, choice and outcome) in each trial type, by averaging firing rates for each neuron across all trials and across time bins in each epoch. Subsequently, we conducted a standard principal component analysis on the 125-dimensional, 5-epoch, 2-trial-type, population responses, using the Statistics and Machine Learning Toolbox of MATLAB (https://www.mathworks.com/help/stats/pca.html).

**Analysis windows.** The analysis window is the peak-encoding period, within each event epoch by task design, based on 300 ms-window sliding regression analysis of normalized firing rates against predictor of interest. Offer 1 analysis window lasted 300 ms after 200 ms of offer 1 onset. Offer 2 analysis window was defined as a 300 ms window around peak encoding of Offer 2 size within Offer 2 presentation epoch. Since after onset of choice epoch, the trial would not precede till subjects successfully make a choice and the decision time varied trial by trial, we defined choice epoch as a 1,000 ms window within choice period. Outcome analysis window was defined as a 400 ms window around peak encoding of outcome size within outcome event epoch. Inter-trial interval was defined as a 1,000 ms epoch following the outcome epoch.

**Data availability.** The data sets generated during the current study are available on the Hayden lab website, http://www.haydenlab.com/, or from the authors on reasonable request. The code generated to do the analyses for the current study is available from the corresponding author on reasonable request.

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

## Acknowledgements

We thank M. Mancarella and M. Castagno for helping with data collection and R. Akaishi for useful comments on the manuscript. This research was supported by a grant to B.Y.H. from the Klingenstein-Simons Foundation and NIH R01 DA037229: Neural Basis of Reward-based Choice.

## Author contributions

B.Y.H. and M.Z.W. designed the experiment, M.Z.W. conducted the experiment and analysed the data, B.Y.H. and M.Z.W. wrote the paper.

## Additional information

**Competing interests:** The authors declare no competing financial interests.

