## [Peer Review File · Nature Communications]

Reviewers' comments:

Reviewer #1 (Remarks to the Author):

The orbitofrontal cortex (OFC) is critical for value-based decision-making. The outcome-reactivation hypothesis states that OFC supports decision-making by reactivating outcome-specific neuronal firing when reward predictive stimuli are presented, and that these reactivation signals are used to guide choices. Different versions of this hypothesis have been put forward, either emphasizing reactivation of outcome-specific signals or reactivation of stimulus-outcome associations that are specifically bound to the current task set (i.e., associative structure). This manuscript reports on a study that uses single unit recordings in monkey OFC (area 13) performing a decision-making task involving two different task sets to test different versions of the outcome-reactivation hypothesis.

Subjects made riskless choices between two sequentially presented offers (offer 1 and offer 2) after which the outcome corresponding to the chosen offer was delivered. Two task-sets were used: On "described" trials, the size of offer 1 was reliably indicated by the color of a visual cue. On "experienced" trials, the size of offer 1 was revealed by delivering the offered amount of water to the subject. The results show that within each of the two task sets, outcome-related activity patterns were reactivated during the offer, such that unit-wise regression slopes relating firing rates to reward size were significantly correlated between the offer and the outcome. In line with this, it was possible to decode outcome size based on activity patterns from the offer and vice versa. In addition, regression slopes coding for outcome size on "experienced" and "described" trials were correlated. However, regression slopes coding for offers on "described" trials did not correlate with regression slopes for outcomes on "experienced" trials. The same was true for regression slopes for offers on "experienced" trials, which did not correlate with regression slopes for outcomes on "described" trials, even though the exact same water reward was delivered in both events.

This pattern of results is partially compatible/incompatible with both reactivation models. There is clear evidence for reactivation of outcome patterns, and these are partially outcome general and partially specific for a given task-set. In fact, the results support a third model which offers a reconciliation for these two seemingly inconsistent sets of results. The model proposed by the authors states that there is a general response to reward outcomes that is shared across task-sets. However, there is also a task-set specific component in the outcome-related activity pattern. Importantly, only the task-set specific component is reactivated during the offer phase.

General comments

The authors address a very important and interesting question. The design is extremely clever and the experiments are executed with great attention to detail. The data analysis is straightforward but elegant, and the results are fully convincing. On top of that, the manuscript is extremely well written - it was a real pleasure to read. I have only a few minor comments and suggestions related to the discussion and the presentation of the findings, which the authors may want to consider.

Specific comments:

1. In the abstract, the authors state that "Others propose that it calls upon a response pattern that is specific to each offer-outcome pair." I think this statement is a little misleading and does not really capture what (I believe) the authors have in mind. The hypothesis is not that each offer-outcome pair has a unique pattern that is reactivated, but that encoding of offer-outcome pairs is specific to the "associative structure" or "task set" in which they are embedded. For instance, in case there were two sets of stimuli describing the same offer size, encoding of offer size in one of these sets would overlap with the coding in the second set, wouldn't it? Similarly, I take it that regression slopes for offer size would be correlated if they were separately estimated on trials with 200 and 75 μL , and on trials with 250 and 100 μL (this could perhaps be tested). Also, decoding would be possible across offers and outcomes if the decoder was trained and tested on different offer subsets (there are probably too few small-size outcome trials to actually test this)? This would be more in line with the findings of Kahnt et al (2010) and Howard et al, (2016) which show that responses to different stimuli predicting the same outcome overlap (besides overlapping responses between stimuli and outcomes), allowing decoding of expected value in response to one set of stimuli if the classifier was trained on a distinct set of stimuli. This, and the results presented here, suggest that reactivation is not so much specific to each offer-outcome pair, but rather to the associative structure (trial-sequence) in which the offer-outcome pair occurs.
2. It would be helpful if the authors could state the proportion of neurons coding offer/outcome size with positive and negative regression slopes.
3. The argument of reactivation is made for the OFC as a whole, yet recordings were only made in area 13. Maybe the authors could include a discussion about whether they think their results are specific (or not) to this particular recoding site.
4. The results convincingly show that there is overlap in how outcome size in the two task sets is encoded. However, cross-decoding seems to work only in one direction, namely training on outcomes on described trials and testing on outcomes on experienced trials. Do the authors have an intuition of why this may be the case? Usually, an asymmetry in cross-decoding indicates increased levels of noise in the training data (i.e., experienced outcomes) resulting in lower accuracy. Could this be related to the fact that subjects had already received the outcome during the offer phase, which could have changed responses during the outcome phase?
5. The authors may want to explain their rationale for using absolute regression coefficients to test for an overlap in the set of neurons involved in coding offers and outcomes (line 185).
6. I believe there is a typo in line 236, it should be Figure 4d, not 4c. The opposite is true in line 256.

Reviewer #2 (Remarks to the Author):

This study concerns the encoding of associative structures in OFC and how neural activity contributes to economic choices. It was designed to compare neural data associated with visual cues or reward itself. The context refers to 2 alternatives: 1) Reward prediction activates neurons like for reward itself: expectation = simulation of reward. Or 2) Distinct activity patterns are produced for every offer-outcome pairing. The authors propose a third alternative or intermediate solution, being that OFC neurons encode both an associative structure specific of offer-outcomes pairs and –simultaneously- a general reward signal. To tests these various alternatives the authors design a task in which neural responses to offers (visual cues associated with reward amounts or actual rewards) and outcomes could be compared.

The approach is interesting and the methodology is appropriate for the purpose of this experiment. Indeed one way to address the question of whether valued stimuli induce reactivation of outcome-related activity is to test decoding algorithms or to compare coding schemata for each event. The authors used multiple methods to validate their results. A particularly clear result is that Described versus Experienced trials seem to evoke different neural states and that encoding of outcome values in both trial types are quite similar. It is convincing although unclear how novel these results are because the structure of trials with different visual cues and even contrasting rewards vs visual cues would naturally be expected to induce different coding schemes.

There are few issues though which in this version reduce the strength of the study. The amount of data is quite small; each neuron is recorded for a large number of trials but the total number of single units is really small (n=60 per monkey) ; this might impact the extent to which one can access the encoding properties of the region (for which we do not have much information by the way; spread of recording sites for instance?). As detailed below, many tests show very weak effects and this could come from the sampling issue.

Another problem resides in the expected outcome of statistical tests and on the interpretation of results. One main concern is which test to use, and what level of significance one must reach to decide that population coding are overlapping sufficiently to call it a common code. This is what the authors refer to as the 'overlap'. For instance, regression analyses on regression coefficients presented in figure 4 are extremely weak, with r^2 around .15 at most; single units with significant encoding of outcome size, offer size etc., are small (9 to 17%), and the number of units with significant encoding of two tested parameters (red points in figure 4) are even smaller (apparently 1 or 2 per figure). How much of these statistics is driven by outliers? Could the authors check for this?

Another sound approach used by the authors is the decoding presented in Figure 5. But as for regressions, decoding scores are very low. The only decoding score that stands out of chance level (actually one could show a standard error for the chance level if calculated from appropriate permutations) is for Outcome vs Offer in Experienced trials which relate to the same physical event. These low scores should not be due to problems with the analytical

procedure because different methods led to similar decoding rates. One could argue that the decoding scores for within-trial data are larger than expected by chance; but could this extra decoding performance be simply due to the autocorrelation statistics of unit activity and to the time constant of neuronal activity in OFC and not to specific properties of encoding values and associative structures?

Minor.

Page 7-8. Encoding of offer 1 in described versus experienced conditions is not different. This paragraph should refer to figure 3c which would suggest otherwise. The statistics related to the data presented in the figure would be informative. In addition and on this same point, I would suggest to include the PSTH and rasters for the 2 conditions (experienced, described) for each neuron in Figure 3a,b.

Page 6: monkeys preferred offer 1 on offer 2, they chose offer_1 44.31% of the time: Am I missing something; the 66% remaining should correspond to choosing offer 2 and hence the preferred option is 2...

Figure 4. The meaning of coloured points should be given in the legend. There are also two types of grey points which are not explained (Fig. 4a, b, and c).

Reviewer #3 (Remarks to the Author):

Wang and Hayden

This is an interesting investigation of the encoding of rewards and the events that predict them in the macaque orbitofrontal cortex (OFC). In essence the authors demonstrate that events that are associated share, or at least partially share, a neural representation. So cues predicting a particular outcome elicit partially similar OFC activity to that elicited by the outcomes themselves. The manuscript is written clearly. I have some minor suggestions for possible additions to the analyses but I think that they will have little impact on interpretation of the results. I have suggested that caveats should perhaps be more clearly acknowledged in some places. Otherwise most points are suggestions for clarifying some arguments and relating the results to some related points in the literature (although I think that the current results are novel and interesting).

1 P5, paragraph 1 In the introduction the authors argue that it is possible that neural representations of outcomes are activated when people or animals see cues or stimuli that predict those outcomes. They argue, however, that it is not clear whether or not this is the case. Several studies are listed at the bottom of page 4/top of page 5 that support this contention. The study by McNamee is cited as supporting the opposing possibility – that outcome representations are not activated by predictive events. However, it seems as if the result reported by McNamee actually suggests that at the time of making a choice the neural activity pattern is predictive of the level of reward that subjects will subsequently receive? Moreover in the case of the study by Tsujimoto and colleagues cited a few lines over, isn't it important to clarify whether the water reward that is used as a cue for the monkey to respond is predictive of further identical rewards or of different rewards?

2 The authors refer to “responses to outcomes” and “outcome responses particularly at the beginning of their report but sometimes later. They mean neural responses to reward outcomes but this can sometimes make the text confusing because they sometimes also talk about responses in another sense: motor responses that animals make to obtain rewards. Would it be helpful to change one of the words so that no confusion is possible? Perhaps confusion could be avoided by talking about neural activation to outcomes or neural representations of rewards?

3 p5, paragraph 2: The authors’ suggestion that reward representations comprise both a specific associative structure predicting a specific reward type and a more general response to reward might be compared with Burke et al., (Nature, 2008) who discuss specific reward effects and general reward-related effects during learning. It might be worth comparing the authors ideas with these related ideas somewhere in the report. Related ideas have also been proposed in the context of a brain structure interconnected with OFC -- the amygdala: general reward information in the central nucleus and outcome specific information in the basolateral nucleus (Killcross and Balleine, TINS, 2006). Perhaps mention in the Discussion?

4 p12, paragraph 3 Wang and Hayden report no correlation between OFC codes for experienced and described offers. This contrasts with the correlations between OFC codes for experienced offers and outcomes and described offers and outcomes. It might be possible to strengthen the conclusions by showing not just that some correlations are significant and others are not significant by showing that some correlations are significantly greater than other correlations. In other words, correlation coefficients might be compared if the authors want to show that some correlations are greater than others. This might be done using a Fisher r to z transformation. However, even if this is not possible the results are very interesting as they stand.

5 Figure 5c suggests that overlap between outcome representations in the two trial types, experienced and described, should not be overemphasized. While there is good decoding from described outcomes to experienced outcomes the reverse does not seem to be the case (second bar of panel 5c). If I have understood the graph correctly then perhaps this could be acknowledged somewhere near the bottom of page 11/top of page 12 where the results relating to the first bar in 5c (which is significant) are discussed.

6 p18, last paragraph – p19 first paragraph. It is argued that the results are consistent with a “cognitive map” theory of OFC function and that “Evidence supporting this theory includes lesions studies showing that OFC is necessary for maintaining a task space representation under the current task demand...”. However, arguably, OFC lesions in macaque monkeys do not support this contention. The Wilson et al., (2014) account emphasizes the importance of representations of hidden task states in OFC and suggests that these are required for reversal tasks. However, Rudebeck and colleagues (Nature Neuroscience, 2013) have argued that precisely such tasks can be performed even in the absence of the OFC. One possibility is that task states depend on an adjacent part of OFC not lesioned by Rudebeck, such as that investigated by Chau and colleagues (Neuron, 2015) but a direct test of such a

possibility needs to be performed.

7 Presumably the "E"s and "D"s in figure 5c refer to experienced and described trials?
Maybe this could be stated explicitly.

Typos:

P28 "Our results are consistent with and the cognitive map theory..." should be "Our results are consistent with the cognitive map theory..."

Reviewer #1 (Remarks to the Authors):

General comments

The authors address a very important and interesting question. The design is extremely clever and the experiments are executed with great attention to detail. The data analysis is straightforward but elegant, and the results are fully convincing. On top of that, the manuscript is extremely well written - it was a real pleasure to read. I have only a few minor comments and suggestions related to the discussion and the presentation of the findings, which the authors may want to consider.

We appreciate these kind comments.

Specific comments:

1. In the abstract, the authors state that “Others propose that it calls upon a response pattern that is specific to each offer-outcome pair.” I think this statement is a little misleading and does not really capture what (I believe) the authors have in mind. The hypothesis is not that each offer-outcome pair has a unique pattern that is reactivated, but that encoding of offer-outcome pairs is specific to the “associative structure” or “task set” in which they are embedded. For instance, in case there were two sets of stimuli describing the same offer size, encoding of offer size in one of these sets would overlap with the coding in the second set, wouldn't it?

We agree with the reviewer: what we have in mind by “offer-outcome pair” actually refers to the specific associative structure. We originally used the less accurate term, “offer-outcome pair”, in the

hope that this term was more readily understandable by a broader audience who might not necessarily be familiar with the term “associative structure” and its related literature. However, since the focus of the current paper is indeed associative structure, and our task was not designed to test whether the different offers (/stimuli) predicting the same reward outcome would share the same encoding format (as was specifically tested in Kahnt et al, 2010 and Howard et al, 2016), we have updated “offer-outcome pair” with “associative structure” in the abstract and the main text accordingly.

Similarly, I take it that regression slopes for offer size would be correlated if they were separately estimated on trials with 200 and 75 μ L, and on trials with 250 and 100 μ L (this could perhaps be tested). Also, decoding would be possible across offers and outcomes if the decoder was trained and tested on different offer subsets (there are probably too few small-size outcome trials to actually test this)? This would be more in line with the findings of Kahnt et al (2010) and Howard et al, (2016) which show that responses to different stimuli predicting the same outcome overlap (besides overlapping responses between stimuli and outcomes), allowing decoding of expected value in response to one set of stimuli if the classifier was trained on a distinct set of stimuli. This, and the results presented here, suggest that reactivation is not so much specific to each offer-outcome pair, but rather to the associative structure (trial-sequence) in which the offer-outcome pair occurs.

We conducted this analysis and found that these two regression slopes were not significantly correlated ($\rho=0.120$, $P= 0.18$). Of course, this positively trending null result is inconclusive and it is likely that the failure to achieve significance reflects that fact that each regression was only performed on a very small number of trials (mean=88 for 200-75 group and mean=90 for 250-100 group). However, we predict that if we had tested with the same method as in Kahnt et al (2010) and Howard et al (2016), according to our model, we indeed would observe that when different visual stimuli were presented with the same associative structure to predict the same reward outcome, the encoding format should be strongly positively correlated. This idea is now made clear in the discussion:

“However, when different visual stimuli but the same associative structure is used to predict the same reward, as in Kahnt (2010), the associative structure specific value encoding during stimuli presentation should resemble a reactivation (Howard, Kahnt, & Gottfried, 2016; Kahnt, Heinzle, Park, & Haynes, 2010).”

2. It would be helpful if the authors could state the proportion of neurons coding offer/outcome size with positive and negative regression slopes.

We agree that such information would be helpful and useful. We have added this information to the results in the revised manuscript, with chi-square tests, as follows:

“During the offer 1 epoch, the size of the described offer affected firing rate in 12% of neurons (n=15/125; linear regression; Figure 3c). This proportion is greater than what would be expected by chance (P=0.002; binomial test). Among these neurons, 53.3% (n=8/15) encoded described offer with positive sign (this proportion is not biased; $\chi^2 < 0.0001$; P=0.5; chi-square test). The size of the experienced offer affected firing rate in the offer 1 epoch in 16.8% of neurons (n=21/125; Figure 3c). This proportion is greater than what would be expected by chance (P<0.001; binomial test). Among experienced offer size-sensitive neurons, 66.7% (n=14/21) encoded experienced offer with positive sign (this proportion is positively biased; $\chi^2 = 3.42$; P=0.032; chi-square test).

The size of the outcome affected firing rate during the outcome epoch in 9.6% of neurons (n=12/125; linear regression; see Methods) in described trials. This proportion is greater than chance (P=0.023; binomial test). Among these neurons, 75.00% (n=9/12) encoded outcomes with negative sign (this proportion is negatively biased; $\chi^2 = 4.17$; P= 0.021; chi-square test). The size of the outcome affected firing rate during the outcome epoch in 12.8% of neurons (n=16/125) in experienced trials. This proportion is greater than chance (P<0.001; binomial test). Among these neurons, 62.50% (n=10/16) encoded outcomes with negative sign (this proportion is not biased; $\chi^2 = 1.13$; P=0.144; chi-square test).”

3. The argument of reactivation is made for the OFC as a whole, yet recordings were only made in area 13. Maybe the authors could include a discussion about whether they think their results are specific (or not) to this particular recoding site.

In the revised manuscript, we now make clear in both abstract and the main text that our results are based on data from Area 13 of OFC. We have also added the following discussion as a speculation about the generalization of this result to other areas of OFC.

“It is important to note that, although OFC lesion in rodents impairs performance in a broad set of tasks that rely on representing the “cognitive map”, such as reinforcer devaluation, reversal learning, and Pavlovian-instrumental (Bradfield, Dezfouli, van Holstein, Chieng, & Balleine, 2015; Wilson, Takahashi, Schoenbaum, & Niv, 2014), the results in monkeys are more heterogeneous. For example, excitotoxic lesion of medial OFC in monkeys impaired performance only in reinforcer devaluation but not reversal learning (Rudebeck, Saunders, Prescott, Chau, & Murray, 2013). One possibility is that reversal learning relies on the adjacent lateral OFC in monkeys (B. K. H. Chau et al., 2015; Walton, Behrens, Buckley, Rudebeck, & Rushworth, 2010). Therefore, it is hard to tell whether our results will generalize to other sub-regions of OFC. Speculatively, these various sub-regions of OFC in monkeys may support representations of different aspects of the associative structure or the cognitive map. This possibility calls for direct test in future research.”

4. The results convincingly show that there is overlap in how outcome size in the two task sets is encoded. However, cross-decoding seems to work only in one direction, namely training on outcomes on described trials and testing on outcomes on experienced trials. Do the authors have an intuition of why this may be the case? Usually, an asymmetry in cross-decoding indicates increased levels of noise in the training data (i.e., experienced outcomes) resulting in lower accuracy. Could this be related to the fact that subjects had already received the outcome during the offer phase, which could have changed responses during the outcome phase?

As the reviewer points out, the asymmetry in cross-decoding is likely attributable to the noisiness in the training data. This is particularly true in our data in the outcome epoch. The fact that subjects made free choices means that they seldom chose and received smaller sized offers (they, not surprisingly, preferred the larger offer). Therefore, some neurons have few trials corresponding to choosing and receiving smaller-sized reward as outcome. This is more so in experienced trials, where subjects were more reward maximizing than in described trials (Figure 2).

We think it is unlikely that the variability during experienced trials was due to the doubled reward. Several factors motivate this view. First, the effect sizes of the offer 1 and outcome responses in experienced trials were not statistically different ($t=0.98$; $P=0.32$). Second, the proportion of neurons tuned for offer 1 and outcome in experienced trials were not significantly different ($\chi^2=0.51$; $P=0.23$; chi-square test). Third, the effect sizes of the outcome responses in

described and experienced trials were not statistically different across trial types ($t=-0.66$; $P=0.51$; paired sample t-test). For these reasons, we suspect the reviewer's first guess is more likely correct.

We have added the following explanation to the Supplementary Note:

“We observed asymmetry in cross-decoding between neural activation states to outcomes in described and in experienced trials. This asymmetry is likely due to a high level of noise in training data (i.e. neural activation to experienced outcomes). The fact that subjects made free choices means that they seldom chose and received smaller sized offers (they, not surprisingly, preferred the larger offer). Therefore, some neurons have few trials corresponding to choosing and receiving smaller-sized reward as outcome. It is more so in experienced trials, where subjects were more reward maximizing than in described trials (Figure 2).”

5. The authors may want to explain their rationale for using absolute regression coefficients to test for an overlap in the set of neurons involved in coding offers and outcomes (line 185).

We have added the following explanation to the main text before reporting the results from this analysis.

“We then tested whether there is an overlap in the set of neurons involved in encoding offer 1 and in encoding outcome. To do so, we used a technique we devised and used for this purpose in earlier studies (Blanchard, Hayden, & Bromberg-Martin, 2015; Strait, Sleezer, & Hayden, 2015). Specifically, we took the absolute value of the two sets of linear regression coefficients (mentioned above) as an index of task participation (i.e. a measure of unsigned coding strength). If the same – or at least a positively overlapping - group of neurons participates in representing the values of offer and outcome, then the absolute value of the regression coefficients for offer and outcome will be positively correlated. Conversely, if there are distinct populations, we will observe a significant negative correlation between these variables. The reason lies in the fact that if there are separable populations, then stronger selectivity for one option implies weaker selectivity for the other one, and will therefore produce a negative correlation. Finally, if there is no special relationship between the populations, and parameter sensitivity is

distributed randomly across the population, we will see no correlation between these variables.”

6. I believe there is a typo in line 236, it should be Figure 4d, not 4c. The opposite is true in line 256.

We thank the reviewer for pointing out these typos and we are sorry for the mixed reference to these figures. We have corrected these typos in the revised manuscript.

Reviewer #2 (Remarks to the Author):

The approach is interesting and the methodology is appropriate for the purpose of this experiment. Indeed one way to address the question of whether valued stimuli induce reactivation of outcome-related activity is to test decoding algorithms or to compare coding schemata for each event. The authors used multiple methods to validate their results. A particularly clear result is that Described versus Experienced trials seem to evoke different neural states and that encoding of outcome values in both trial types are quite similar. It is convincing although unclear how novel these results are because the structure of trials with different visual cues and even contrasting rewards vs visual cues would naturally be expected to induce different coding schemes.

In a series of critical empirical and review papers, OFC was demonstrated, via lesion studies, to be necessary for using knowledge about the associative structure under current demands to guide goal-directed behavior in both decision-making and learning (e.g. Bradfield et al., 2015; Jones et al., 2012; Stalnaker, Cooch, & Schoenbaum, 2015; Wikenheiser & Schoenbaum, 2016; Wilson et al., 2014). Although crucial, these lesion results inevitably lack the ability to shed light on *how* the OFC represents associative structures and *how* this representation is involved in guiding goal-directed behaviors such as reward-based decision making. To our knowledge, our study is the first to tackle these questions head-on with single-unit recordings. We do so by using a goal-directed, reward-based decision-making task, and by controlling for reward identity and reward value while varying only the underlying associative structures. Our finding that during prospective evaluation in a economic choice task, OFC uses mirroring but separated population activation states to encode offers presented with different associative structures but predicting the same rewards, suggest that OFC emphasizes how the reward-predicting events will unfold and how to obtain the reward during pre-choice phases. Subsequently, OFC uses both associative structure specific

and reward general encodings during post-choice phase (reward outcome delivery), suggesting that it has the potential of binding the general reward signal with the reward predicting event sequence and possibly provide this multiplexed learning signal to update or reinforce the current associative structure.

Associative structure, however, is a broad term, which includes “how to obtain the expected reward, its unique form and features, and current value” (Jones et al., 2012). As the reviewer points out, the modality difference between described (visual) and experienced (gustatory and visceral) offers might naturally lead to their difference in coding formats. Although we do not think this is a drawback of the current task design, we do acknowledge that our task could not delineate whether different aspects of the associative structure (e.g. modality/unique form and features vs. how to obtain the expected reward) would have differential effects on neural representation or on its effectiveness in guiding choice behavior. But this might be an intriguing venue for future research.

There are few issues though which in this version reduce the strength of the study.

The amount of data is quite small; each neuron is recorded for a large number of trials but the total number of single units is really small (n=60 per monkey); this might impact the extent to which one can access the encoding properties of the region (for which we do not have much information by the way; spread of recording sites for instance?). As detailed below, many tests show very weak effects and this could come from the sampling issue.

The reviewer expresses concern about the spread of the recording sites. All of our recordings are confined in Area 13 of OFC. The targeted area expands along the coronal planes situated between 28.65 mm and 33.60 mm rostral to the interaural plane with varying depth (as shown in the figures below). On our recording grid (18-by-18 NAN grid), our data were collected from 16 out of 21 holes covering Area 13 of OFC with various depths for subject H and 15 out of 28 holes for subject B. Therefore, we believe that sample is quite well spread within Area 13 of OFC.

Most rostral recording site:

Most caudal recording site:

The reviewer also raises a concern about our sample size ($n=125$, 60 in subject B and 65 in subject H). We would point out, first, that the sample size in our study is not particularly small. For example, the following recent studies use sample sizes from OFC equivalent to ($n=126$ in one study) or smaller than ours:

- [112 and 70 neurons in two tasks] Bouret & Richmond, 2010;
- [85 neurons] Chang, Gariépy, & Platt, 2012;
- [84 neurons] Grattan & Glimcher, 2014;
- [103 neurons] Ifuku, Ohgushi, Ito, & Ogawa, 2002;
- [115 neurons] Kravitz & Peoples, 2008;
- [77 neurons] Lara, Kennerley, & Wallis, 2009;
- [86 neurons] McCracken, & Grace, 2007;
- [60 neurons] Pritchard, Nedderman, Edwards, Petticoffer, Schwartz, & Scott, 2008;
- [80 neurons] Simmons & Richmond, 2008;
- [126 neurons] Wallis, Anderson, & Miller, 2001.

We chose our number of cells based on our own previous studies of this region, which used similar numbers of cells (e.g. Blanchard et al., 2015). We determined our sample size before collecting data and did not “peek” at the data during collection. This approach is

designed to minimize the danger of p-hacking (erroneously rejecting null hypothesis based on questionable statistical methods when the null hypothesis is true). Another motivating factor is our institutional IACUC, which requires us for ethical reasons to collect enough data to test our hypothesis, but not more.

There are two dangers associated with not enough data: (1) low statistical power, i.e. not able to detect an effect, or, accepting null hypothesis when null hypothesis is false; (2) vulnerability to influence of potential outliers.

Problem (1) is a possible issue only when we report and draw conclusion from a null result. We do this once, when we report that the coding formats of described and experienced offers do not overlap ($r=0.02$; $P=0.828$; Spearman's correlation; Figure 4e). In this case, we dealt with the dangers of possible low power directly by including both a power analysis and a permutation test.

“This lack of correlation was not due to lack of power or spurious distribution of the coefficients. We performed a power analysis and a permutation test (Figure 4f; see Methods for details) and both analyses suggested that given our sample size, if a significant correlation truly existed, we would have observed a correlation coefficient (effect size) between -1 to -0.19 or 0.19 to 1, instead of 0.02.”

Problem (2) is raised by the reviewer, and we dealt with this problem in two ways.

First, we now include a new direct test for outliers. Specifically, we tested whether any regression coefficients used in these ensemble analyses qualified as outliers with a Cook's D test. Cook's D test measures global influence that combines leverage and discrepancy. Data points with Cook's D ≥ 1 is usually considered as outliers (Cohen, Cohen, West, & Aiken, 2013). We measured Cook's D for each pair of the regression coefficient sets used for ensemble analyses. As shown below, each dot represent a neuron, and no data point showed Cook's D ≥ 1 . (or even > 0.5 for that matter; these figures now appear in the Supplementary Fig. 2).

Second, outliers could be a potential problem in correlation tests. Thus, for all the ensemble analyses correlating two sets of regression coefficients, we intentionally chose Spearman's over Pearson correlation because it is more robust to outliers.

SF 2 c. Cook's D of Regression Coefficients between Described and Experienced Offer

SF 2 d. Cook's D of Regression Coefficients between Described and Experienced Outcome

SF 2 c. Cook's D between Regression Coefficients for Described and Experienced Offer

SF 2 d. Cook's D between Regression Coefficients for Described and Experienced Outcome

Another problem resides in the expected outcome of statistical tests and on the interpretation of results. One main concern is which test to use, and what level of significance one must reach to decide that population coding are overlapping sufficiently to call it a common code. This is what the authors refer to as the 'overlap'. For instance, regression analyses on regression coefficients presented in figure 4 are extremely weak, with r^2 around .15 at most; single units with significant encoding of outcome size, offer size etc., are small (9 to 17%), and the number of units with significant encoding of two tested parameters (red points in figure 4) are even smaller (apparently 1 or 2 per figure). How much of these statistics is driven by outliers? Could the authors check for this?

The reviewer raises an important point on effect size. It is correct that the percent of neurons with significant effects is small, but we believe that they are typical of neurons in this region (even in studies with much larger sample sizes). Two factors make our numbers appear smaller than in some other published studies. First, we do not pre-select cells based on their firing rate properties; nor do we analyze only the subsample of neurons showing significant

task selectivity. We analyze all neurons. Second, we do not explore multiple epochs and take effects occurring in any epochs as meaningful. Instead, we focus on a single epoch for each hypothesis, chosen a priori. These decisions reduce the risk of p-hacking, of “voodoo correlations”, and of inflated effect size, and thus, we think, provide more accurate measures of true effect sizes.

As for the reviewer’s question about outliers, this is an important problem, and one that needs to be addressed directly when dealing with data like this. As noted above, we now include Cook’s D test to demonstrate that outliers are unlikely to be a major problem in our data, and we use Spearman’s correlations to reduce the potential effects of outliers in our correlation tests.

Another sound approach used by the authors is the decoding presented in Figure 5. But as for regressions, decoding scores are very low. The only decoding score that stands out of chance level (actually one could show a standard error for the chance level if calculated from appropriate permutations) is for Outcome vs Offer in Experienced trials which relate to the same physical event. These low scores should not be due to problems with the analytical procedure because different methods led to similar decoding rates. One could argue that the decoding scores for within-trial data are larger than expected by chance; but could this extra decoding performance be simply due to the autocorrelation statistics of unit activity and to the time constant of neuronal activity in OFC and not to specific properties of encoding values and associative structures?

We think that the relatively low decoding accuracy was mainly contributed by the fact that single neurons are mostly noisy information channels. Moreover, the fact that subjects made free choices means that they seldom chose and received smaller sized offers (they, not surprisingly, preferred the larger offer). Therefore, some neurons have few trials corresponding to choosing and receiving smaller-sized reward as outcome. Therefore, error rates might be inflated for decoder trained with these noisy training data (also see Supplementary Fig. 4).

The second point is that, within each trial type, the higher decoding accuracy between population activation states (the coding pattern) could be simply due to the autocorrelation statistics but not the task manipulations. We agree that autocorrelation is a possibility in theory, but is unlikely in our dataset. This is because we fully interleaved trial types, and so any autocorrelation would create false positives for all comparisons. However, we don’t see this result in our data. Let us take described trials for example. In the manuscript,

analysis on encoding pattern overlap between described offer 1 and its corresponding reward outcome was based on trials where offer 1 was chosen ($r=0.27$; $P=0.003$; Spearman's correlation). If this significant correlation were mainly caused by autocorrelation then we would expect on trials where offer 1 was rejected (i.e. offer 2 was chosen), due to the shared time course, we still would see a significant overlap between coding for offer 1 and outcome, even though in these trials, outcome corresponded to offer 2 instead of offer 1. However, when tested, this correlation is not significant ($r=0.10$; $P=0.251$; Spearman's correlation). Moreover, when comparing the effect size of these two correlation coefficients ($r=0.27$ in offer 1 chosen trials and $r=0.10$ in offer 1 rejected trials), the correlation coefficient in offer 1 chosen trials is significantly larger than that in offer 1 rejected trials ($z\text{-value}= 1.95$, $p=0.026$; Fisher's transformation test). These control analyses demonstrate that autocorrelation in firing rates does not provide much of a confound in this dataset.

Minor.

Page 7-8. Encoding of offer 1 in described versus experienced conditions is not different. This paragraph should refer to figure 3c which would suggest otherwise. The statistics related to the data presented in the figure would be informative.

We have made the changes according to the reviewer's suggestion.

3 c. Proportion of Neurons Modulated by Offer 1 Size

In addition and on this same point, I would suggest to include the PSTH and rasters for the 2 conditions (experienced, described) for each neuron in Figure 3a,b.

Due to the fact that example neuron #69 is tuned to described but not experienced offer 1 and vice versa for #123, we have added the figures to Supplementary Figure 1.

SF 1 a. Modulation by Experienced Offer 1 for Cell # 69

SF 1 a. Modulation by Described Offer 1 for Cell # 123

Page 6: monkeys preferred offer 1 on offer 2, they chose offer_1 44.31% of the time: Am I missing something; the 66% remaining should correspond to choosing offer 2 and hence the preferred option is 2...

We thank the reviewer for pointing out this wording mistake. We meant that subjects chose offer 1 more often than expected by optimal strategy. We have updated the revised manuscript accordingly.

“Subjects chose offer 1 more often than expected by optimal strategy: they chose it 44.31% of the time (even though its value was matched to or better than offer 2 only 40% of the time; $\chi^2=120.77$; $P<0.001$; chi-square test).”

Figure 4. The meaning of coloured points should be given in the legend. There are also two types of grey points which are not explained (Fig. 4a, b, and c).

We have updated the figure legend and caption according to the reviewer's suggestions.

Reviewer #3 (Remarks to the Author):

Wang and Hayden

This is an interesting investigation of the encoding of rewards and the events that predict them in the macaque orbitofrontal cortex (OFC). In essence the authors demonstrate that events that are associated share, or at least partially share, a neural representation. So cues predicting a particular outcome elicit partially similar OFC activity to that elicited by the outcomes themselves. The manuscript is written clearly. I have some minor suggestions for possible additions to the analyses but I think that they will have little impact on interpretation of the results. I have suggested that caveats should perhaps be more clearly acknowledged in some places. Otherwise most points are suggestions for clarifying some arguments and relating the results to some related points in the literature (although I think that the current results are novel and interesting).

1 P5, paragraph 1 In the introduction the authors argue that it is possible that neural representations of outcomes are activated when people or animals see cues or stimuli that predict those outcomes. They argue, however, that it is not clear whether or not this is the case. Several studies are listed at the bottom of page 4/top of page 5 that support this contention. The study by McNamee is cited as supporting the opposing possibility – that outcome representations are not activated by predictive events. However, it seems as if the result reported by McNamee actually suggests that at the time of making a choice the neural activity pattern is predictive of the level of reward that subjects will subsequently receive?

The reviewer is correct that the level of reward was decodable during presentation of the offer stimuli (prior to executing a chosen response). In McNamee et al (2015), each trial contained a unique associative event sequence: a stimulus, a freely chosen action, and an outcome cue. Each unique combination of the three events led to either a high or low reward state. After seeing the stimulus, participants had to choose between one of the two actions in order to carry out the sequence that would lead to the desired reward state. Successful decoding of the reward state during both stimulus

and action epochs from central OFC suggests that OFC correctly predicted the associative event sequence before and during execution of the action that could lead to the decoded reward state. Therefore, we think that, in this study, prospective reactivation of outcome response is imbedded in representing unique associative structures. The difference between their and our studies, though, is that we specifically tested whether OFC differentiate different associative structures when they lead to the same reward outcome.

Moreover, associative structure is defined as “how to obtain the expected reward, its unique form and features, and current value” (Jones et al., 2012). Therefore, the representation of associative structure could include both associative event sequence (how to obtain the expected reward) and the value. Indeed, we report that during offer epoch, value was only represented in associative structure specific format but not reward general format, and thus was potentially represented along with knowledge such as “how to obtain the expected reward”, in order to guide choice behavior.

We have now updated the manuscript with the following paragraph for better clarity: **“In one task, each unique associative event sequence (a visual stimulus, an freely chosen action, and an outcome cue) led to a high or low reward state. After seeing the visual stimulus, participants freely chose and performed one of two actions to complete the sequence that led to the desired reward. The reward states predicted by each sequence were decodable during stimulus presentation and action execution in human central OFC, suggesting that the reward information was represented based on the unique underlying associative structure (McNamee, Liljeholm, Zika, & O’Doherty, 2015).”**

Moreover in the case of the study by Tsujimoto and colleagues cited a few lines lower, isn’t it important to clarify whether the water reward that is used as a cue for the monkey to response is predictive of further identical rewards or of different rewards?

We agree with the reviewer. In this cited study, the same sized water reward (0.2 ml) was delivered in two forms: either a single drop of 0.2 ml, or two drops of 0.1 ml. As an instruction, one form was used to instruct the subjects to choose one of the visual targets chosen on the last trial (stay strategy), and the other form was used to instruct the subjects to choose one of the visual targets not chosen on the last trial (switch strategy). As a reward outcome / feedback, both forms of delivery informed subjects of a correct choice

according to the cued strategy, but the two forms were randomly delivered as feedbacks. Therefore, the instruction predicted a correct strategy, and upon execution of the correct strategy, the reward of an identical size was then delivered as a feedback.

We have rephrased this sentence for better clarity as follows:

“Tsujiimoto and colleagues (Tsujiimoto, Genovesio, & Wise, 2012) showed that distinct subsets of macaque OFC neurons encoded the water reward of equal size when it was presented via two routes, as an instruction for choice strategy (stay/switch) versus as a feedback for correct execution of a choice strategy (presumably reflecting distinct associative structures).”

2 The authors refer to “responses to outcomes” and “outcome responses particularly at the beginning of their report but sometimes later. They mean neural responses to reward outcomes but this can sometimes make the text confusing because they sometimes also talk about responses in another sense: motor responses that animals make to obtain rewards. Would it be helpful to change one of the words so that no confusion is possible? Perhaps confusion could be avoided by talking about neural activation to outcomes or neural representations of rewards?

We thank the reviewer for pointing out the ambiguity of these phrases. We now consistently use the phrase “neural activation to outcomes” throughout the revised manuscript.

3 p5, paragraph 2: The authors’ suggestion that reward representations comprise both a specific associative structure predicting a specific reward type and a more general response to reward might be compared with Burke et al., (Nature, 2008) who discuss specific reward effects and general reward-related effects during learning. It might be worth comparing the authors ideas with these related ideas somewhere in the report. Related ideas have also been proposed in the context of a brain structure interconnected with OFC -- the amygdala: general reward information in the central nucleus and outcome specific information in the basolateral nucleus (Killcross and Balleine, TINS, 2006). Perhaps mention in the Discussion?

We thank the reviewer for pointing us towards this important comparison and interpretation about our results. We have now added the following paragraph to the discussion section.

“Relatedly, outcome specific and outcome general reward representation are double dissociable, both behaviorally and

neurally (Balleine & Killcross, 2006; Cardinal, Parkinson, Hall, & Everitt, 2002). For example, OFC lesions specifically impair outcome specific reward value representation and abolish its effect on later blocking and devaluation tests (Burke, Franz, Miller, & Schoenbaum, 2008). Although our task could not directly test this aspect, a generalization from our results suggests that outcome-specific reward representation would be represented as part of the associative structure specific representation during both offer and reward outcome epochs. For more discussion on these subjects, see (Burke et al., 2008; Howard, Gottfried, Tobler, & Kahnt, 2015; Klein-Flügge, Barron, Brodersen, Dolan, & Behrens, 2013; McNamee, Rangel, & O'Doherty, 2013).”

4 p12, paragraph 3 Wang and Hayden report no correlation between OFC codes for experienced and described offers. This contrasts with the correlations between OFC codes for experienced offers and outcomes and described offers and outcomes. It might be possible to strengthen the conclusions by showing not just that some correlations are significant and others are not significant by showing that some correlations are significantly greater than other correlations. In other words, correlation coefficients might be compared if the authors want to show that some correlations are greater than others. This might be done using a Fisher r to z transformation. However, even if this is not possible the results are very interesting as they stand.

The reviewer's point is a good one; there is an opportunity to strengthen the results. We now do so by including results of the recommended analyses in our revised manuscript.

“Moreover, in comparison, correlation coefficient between regression coefficients for described offers and experienced offers is significantly smaller than that between described and experienced outcomes (z -value=2.25, $P=0.012$, Fisher’s Transformation Test). The similar effect was observed in comparison to correlation coefficient between regression coefficients for described offers and outcomes (z -value=2.84, $P=0.002$, Fisher’s Transformation Test) and that between experienced offers and outcomes (z -value=3.94, $P<0.001$, Fisher’s Transformation Test).”

5 Figure 5c suggests that overlap between outcome representations in the two trial types, experienced and described, should not be overemphasized. While there is good decoding from described outcomes to experienced outcomes the reverse does not seem to be the case (second bar of panel 5c). If I have understood the graph correctly then perhaps this could be acknowledged

somewhere near the bottom of page 11/top of page 12 where the results relating to the first bar in 5c (which is significant) are discussed.

We have added the following sentences to this paragraph in the main text and possible explanations in the supplement.

“A decoder trained on activation states for outcome in experienced trials, however, could not significantly decode activation states for outcome in described trials (22.04%; $\chi^2=0.67$; $P=0.21$; chi-square test; Figure 5C). We suspect that high noise in training data contributed to this asymmetry in decoding (see Supplementary Note 1).”

Supplementary Note 1:

“We observed asymmetry in cross-decoding between neural activation states to outcomes in described and in experienced trials. This asymmetry is likely due to a high level of noise in training data (i.e. neural activation to experienced outcomes). The fact that subjects made free choices means that they seldom chose and received smaller sized offers (they, not surprisingly, preferred the larger offer). Therefore, some neurons have few trials corresponding to choosing and receiving smaller-sized reward as outcome. It is more so in experienced trials, where subjects were more reward maximizing than in described trials (Figure 2).”

6 p18, last paragraph – p19 first paragraph. It is argued that the results are consistent with a “cognitive map” theory of OFC function and that “Evidence supporting this theory includes lesions studies showing that OFC is necessary for maintaining a task space representation under the current task demand...”. However, arguably, OFC lesions in macaque monkeys do not support this contention. The Wilson et al., (2014) account emphasizes the importance of representations of hidden task states in OFC and suggests that these are required for reversal tasks. However, Rudebeck and colleagues (Nature Neuroscience, 2013) have argued that precisely such tasks can be performed even in the absence of the OFC. One possibility is that task states depend on an adjacent part of OFC not lesioned by Rudebeck, such as that investigated by Chau and colleagues (Neuron, 2015) but a direct test of such a possibility needs to be performed.

We agree with the reviewer. To make this point clear, we have added the following section to discussion.

“It is important to note that, although OFC lesion in rodents impairs performance in a broad set of tasks that rely on “cognitive map” representation, such as reinforcer devaluation, reversal learning, and Pavlovian-instrumental transfer (Bradfield et al., 2015; Wilson et al., 2014), the results in monkeys are more heterogeneous. For example, excitotoxic lesion of medial OFC in monkeys impaired performance only in reinforcer devaluation but not reversal learning (Rudebeck et al., 2013). One possibility is that reversal learning relies on the adjacent lateral OFC in monkeys (Chau et al., 2015; Walton et al., 2010). Therefore, it is hard to tell whether our results will generalize to other sub-regions of OFC. Speculatively, these various sub-regions of OFC in monkeys may support representations of different aspects of the associative structure or the cognitive map. This possibility calls for direct test in future research.”

7 Presumably the “E”s and “D”s in figure 5c refer to experienced and described trials? Maybe this could be stated explicitly.

The reviewer is correct. We have now explained and emphasized the meaning of “D”s and “E”s in the figure caption as referring to described and experienced trials respectively.

Typos:

P28“Our results are consistent with and the cognitive map theory...” should be “Our results are consistent with the cognitive map theory...”

We thank the reviewer for pointing out the typo. We have corrected it in the revised manuscript.

References

- Balleine, B. W., & Killcross, S. (2006). Parallel incentive processing: an integrated view of amygdala function. *Trends in Neurosciences*, *29*(5), 272–279. <http://doi.org/10.1016/j.tins.2006.03.002>
- Blanchard, T. C., Hayden, B. Y., & Bromberg-Martin, E. S. (2015). Orbitofrontal cortex uses distinct codes for different choice attributes in decisions motivated by curiosity. *Neuron*, *85*(3), 602–614. <http://doi.org/10.1016/j.neuron.2014.12.050>
- Bouret, S., & Richmond, B. J. (2010). Ventromedial and orbital prefrontal neurons differentially encode internally and externally driven motivational values in monkeys. *The Journal of Neuroscience : the Official Journal of the Society for Neuroscience*, *30*(25), 8591–8601. <http://doi.org/10.1523/JNEUROSCI.0049-10.2010>
- Bradfield, L. A., Dezfouli, A., van Holstein, M., Chieng, B., & Balleine, B. W. (2015). Medial Orbitofrontal Cortex Mediates Outcome Retrieval in Partially Observable Task Situations. *Neuron*, *88*(6), 1268–1280. <http://doi.org/10.1016/j.neuron.2015.10.044>
- Burke, K. A., Franz, T. M., Miller, D. N., & Schoenbaum, G. (2008). The role of the orbitofrontal cortex in the pursuit of happiness and more specific rewards. *Nature*, *454*(7202), 340–344. <http://doi.org/10.1038/nature06993>
- Cardinal, R. N., Parkinson, J. A., Hall, J., & Everitt, B. J. (2002). Emotion and motivation: the role of the amygdala, ventral striatum, and prefrontal cortex. *Neuroscience and Biobehavioral Reviews*, *26*(3), 321–352. [http://doi.org/10.1016/S0149-7634\(02\)00007-6](http://doi.org/10.1016/S0149-7634(02)00007-6)
- Chau, B. K. H., Sallet, J., Papageorgiou, G. K., Noonan, M. P., Bell, A. H., Walton, M. E., & Rushworth, M. F. S. (2015). Contrasting Roles for Orbitofrontal Cortex and Amygdala in Credit Assignment and Learning in Macaques. *Neuron*, *87*(5), 1106–1118. <http://doi.org/10.1016/j.neuron.2015.08.018>
- Chang, S. W. C., Gariépy, J.-F., & Platt, M. L. (2012). Neuronal reference frames for social decisions in primate frontal cortex. *Nature Neuroscience*, *16*(2), 243–250. <http://doi.org/10.1038/nn.3287>
- Cohen, J., Cohen, P., West, S. G., & Aiken, L. S. (2013). Applied multiple regression/correlation analysis for the behavioral sciences.
- Grattan, L. E., & Glimcher, P. W. (2014). Absence of spatial tuning in the orbitofrontal cortex. *PLoS ONE*, *9*(11), e112750. <http://doi.org/10.1371/journal.pone.0112750>
- Howard, J. D., Gottfried, J. A., Tobler, P. N., & Kahnt, T. (2015). Identity-specific coding of future rewards in the human orbitofrontal cortex. *Proceedings of the National Academy of Sciences of the United States of America*, *112*(16), 5195–5200. <http://doi.org/10.1073/pnas.1503550112>
- Howard, J. D., Kahnt, T., & Gottfried, J. A. (2016). Converging prefrontal pathways support associative and perceptual features of conditioned stimuli. *Nature Communications*, *7*, 11546. <http://doi.org/10.1038/ncomms11546>
- Ifuku, H., Ohgushi, M., Ito, S., & Ogawa, H. (2002). Neurons associated with behavioral context errors in the primary and higher-order gustatory cortices in the monkey. *Neuroscience Letters*, *319*(2), 121–123.
- Jones, J. L., Esber, G. R., McDannald, M. A., Gruber, A. J., Hernandez, A., Mirenzi, A.,

- & Schoenbaum, G. (2012). Orbitofrontal cortex supports behavior and learning using inferred but not cached values. *Science*, 338(6109), 953–956. <http://doi.org/10.1126/science.1227489>
- Kahnt, T., Heinzle, J., Park, S. Q., & Haynes, J.-D. (2010). The neural code of reward anticipation in human orbitofrontal cortex. *Proceedings of the National Academy of Sciences of the United States of America*, 107(13), 6010–6015. <http://doi.org/10.1073/pnas.0912838107>
- Klein-Flügge, M. C., Barron, H. C., Brodersen, K. H., Dolan, R. J., & Behrens, T. E. J. (2013). Segregated encoding of reward-identity and stimulus-reward associations in human orbitofrontal cortex. *The Journal of Neuroscience : the Official Journal of the Society for Neuroscience*, 33(7), 3202–3211. <http://doi.org/10.1523/JNEUROSCI.2532-12.2013>
- Kravitz, A. V., & Peoples, L. L. (2008). Background firing rates of orbitofrontal neurons reflect specific characteristics of operant sessions and modulate phasic responses to reward-associated cues and behavior. *The Journal of Neuroscience : the Official Journal of the Society for Neuroscience*, 28(4), 1009–1018. <http://doi.org/10.1523/JNEUROSCI.4344-07.2008>
- Lara, A. H., Kennerley, S. W., & Wallis, J. D. (2009). Encoding of Gustatory Working Memory by Orbitofrontal Neurons, 29(3), 765–774. <http://doi.org/10.1523/JNEUROSCI.4637-08.2009>
- McCracken, C. B., & Grace, A. A. (2007). High-Frequency Deep Brain Stimulation of the Nucleus Accumbens Region Suppresses Neuronal Activity and Selectively Modulates Afferent Drive in Rat Orbitofrontal Cortex In Vivo. *Journal of Neuroscience*, 27(46), 12601–12610. <http://doi.org/10.1523/JNEUROSCI.3750-07.2007>
- McNamee, D., Liljeholm, M., Zika, O., & O'Doherty, J. P. (2015). Characterizing the associative content of brain structures involved in habitual and goal-directed actions in humans: a multivariate fMRI study. *The Journal of Neuroscience : the Official Journal of the Society for Neuroscience*, 35(9), 3764–3771. <http://doi.org/10.1523/JNEUROSCI.4677-14.2015>
- McNamee, D., Rangel, A., & O'Doherty, J. P. (2013). Category-dependent and category-independent goal-value codes in human ventromedial prefrontal cortex. *Nature Publishing Group*, 16(4), 479–485. <http://doi.org/10.1038/nn.3337>
- Pritchard, T. C., Nedderman, E. N., Edwards, E. M., Petticoffer, A. C., Schwartz, G. J., & Scott, T. R. (2008). Satiety-responsive neurons in the medial orbitofrontal cortex of the macaque. *Behavioral Neuroscience*, 122(1), 174–182. <http://doi.org/10.1037/0735-7044.122.1.174>
- Rudebeck, P. H., Saunders, R. C., Prescott, A. T., Chau, L. S., & Murray, E. A. (2013). Prefrontal mechanisms of behavioral flexibility, emotion regulation and value updating. *Nature Neuroscience*, 16(8), 1140–1145. <http://doi.org/10.1038/nn.3440>
- Simmons, J. M., & Richmond, B. J. (2008). Dynamic changes in representations of preceding and upcoming reward in monkey orbitofrontal cortex. *Cerebral Cortex*, 18(1), 93–103. <http://doi.org/10.1093/cercor/bhm034>
- Stalnaker, T. A., Cooch, N. K., & Schoenbaum, G. (2015). What the orbitofrontal cortex does not do. *Nature Neuroscience*, 18(5), 620–627. <http://doi.org/10.1038/nn.3982>
- Strait, C. E., Sleezer, B. J., & Hayden, B. Y. (2015). Signatures of Value Comparison in

- Ventral Striatum Neurons. *PLoS Biology*, *13*(6), e1002173–22.
<http://doi.org/10.1371/journal.pbio.1002173>
- Tsujimoto, S., Genovesio, A., & Wise, S. P. (2012). Neuronal activity during a cued strategy task: comparison of dorsolateral, orbital, and polar prefrontal cortex. *The Journal of Neuroscience : the Official Journal of the Society for Neuroscience*, *32*(32), 11017–11031. <http://doi.org/10.1523/JNEUROSCI.1230-12.2012>
- Wallis, J. D., Anderson, K. C., & Miller, E. K. (2001). Single neurons in prefrontal cortex encode abstract rules. *Nature*, *411*(6840), 953–956. <http://doi.org/10.1038/35082081>
- Walton, M. E., Behrens, T. E. J., Buckley, M. J., Rudebeck, P. H., & Rushworth, M. F. S. (2010). Separable learning systems in the macaque brain and the role of orbitofrontal cortex in contingent learning. *Neuron*, *65*(6), 927–939.
<http://doi.org/10.1016/j.neuron.2010.02.027>
- Wikenheiser, A. M., & Schoenbaum, G. (2016). Over the river, through the woods: cognitive maps in the hippocampus and orbitofrontal cortex. *Nature Reviews. Neuroscience*, *17*(8), 1–11. <http://doi.org/10.1038/nrn.2016.56>
- Wilson, R. C., Takahashi, Y. K., Schoenbaum, G., & Niv, Y. (2014). Orbitofrontal cortex as a cognitive map of task space. *Neuron*, *81*(2), 267–279.
<http://doi.org/10.1016/j.neuron.2013.11.005>

REVIEWERS' COMMENTS:

Reviewer #1 (Remarks to the Author):

The authors have fully addressed my initial points. I have no additional comments on this outstanding study.

Reviewer #2 (Remarks to the Author):

The authors responded to all comments including statistical arguments. My comments about the sample size was not in absolute terms but concerned whether it was sufficient for this particular study and its specific analyses. The authors responded that they define a priori their sample size, which I can understand, although I don't quite see which method they would choose to do this. Finally, I don't think limited sample sizes and p-hacking are the same issues as the authors response might suggest.

I suggest the anatomical data regarding recording sites are made accessible to readers.

Reviewer #3 (Remarks to the Author):

Reviewer 3: The authors have addressed all of my previous comments. I continue to think that this is an interesting study of an important aspect of behaviour.

Reviewer #1 (Remarks to the Authors):

The authors have fully addressed my initial points. I have no additional comments on this outstanding study.

Reviewer #2 (Remarks to the Author):

The authors responded to all comments including statistical arguments. My comments about the sample size was not in absolute terms but concerned whether it was sufficient for this particular study and its specific analyses. The authors responded that they define a priori their sample size, which I can understand, although I don't quite see which method they would choose to do this. Finally, I don't think limited sample sizes and p-hacking are the same issues as the authors response might suggest.

We defined a priori our sample size of the current study with a power analysis. Specifically, power analysis estimates the minimum sample size required to detect an effect of a given size with a certain degree of confidence (significance level, i.e. probability of Type I error, and, power, i.e. 1 minus probability of Type II error). To estimate the effect size, we used the mean effect size of a previous study from our lab that recorded in the same region (Area 13 of OFC) and conducted the same ensemble analysis as in the current study (Blanchard, Hayden, & Bromberg-Martin, 2015). In this previous study, mean effect size of significant correlations between two sets of regression coefficients is $r = 0.386$ (effect size of all significant correlations reported in the paper: 0.68, 0.33, 0.41, 0.31, and 0.2). We used 0.05 as significance level and 0.85 as power. Entering these parameters into the power analysis, the minimum sample size required to detect an effect size of 0.386 with significance level 0.05 and power 0.85 is $n=57$. In order to replicate the same effect in two animals, our goal was to collect at least 57 neurons from each animal. Eventually, we collected 65 and 60 neurons from each animal, respectively.

I suggest the anatomical data regarding recording sites are made accessible to readers.

We agree with the reviewer and the related anatomical data and figures now appear in Supplementary Figure 1.

Supplementary Figure 1.

SF 1 a. Most rostral recording site:

SF 1 b. Most caudal recording site:

Reviewer #3 (Remarks to the Author):

The authors have addressed all of my previous comments. I continue to think that this is an interesting study of an important aspect of behaviour.

References

Blanchard, T. C., Hayden, B. Y., & Bromberg-Martin, E. S. (2015). Orbitofrontal cortex uses distinct codes for different choice attributes in decisions motivated by curiosity. *Neuron*, 85(3), 602–614. <http://doi.org/10.1016/j.neuron.2014.12.050>